



# Disentangling different moisture transport pathways over the eastern subtropical North Atlantic using multi-platform isotope observations and high-resolution numerical modelling

Fabienne Dahinden[1], Franziska Aemisegger[1], Heini Wernli[1], Matthias Schneider[2], Christopher J. Diekmann[2], Benjamin Ertl[2], Peter Knippertz[2], Martin Werner[3], and Stephan Pfahl[4]

[1]Institute for Atmospheric and Climate Science, ETH Zurich, Zurich, Switzerland
[2]Institute of Meteorology and Climate Research, Karlsruhe Institute of Technology, Karlsruhe, Germany
[3]Alfred Wegener Institute, Helmholtz Centre for Polar and Marine Research, Bremerhaven, Germany
[4]Institute for Meteorology, Freie Universität Berlin, Berlin, Germany

*Correspondence to*: Fabienne Dahinden (fabienne.dahinden@env.ethz.ch)

**Abstract.** Due to its dryness, the subtropical free troposphere plays a critical role in the radiative balance of the Earth's climate system. But the complex interactions of the dynamical and physical processes controlling the variability in the moisture budget of this sensitive region of the subtropical atmosphere are still not fully understood. Stable water isotopes can provide important information about several of the latter processes, namely subsidence drying, turbulent mixing, dry and moist convective moistening. In this study, we use high-resolution simulations of the isotope-enabled version of the regional weather and climate prediction model of the Consortium for Small-Scale Modelling (COSMO$_{iso}$) to investigate predominant moisture transport pathways in the Canary Islands region in the eastern subtropical North Atlantic. Comparison of the simulated isotope signals with multi-platform isotope observations (aircraft-based in situ measurements, ground-based and space-based remote sensing observations) from a field campaign in summer 2013 shows that COSMO$_{iso}$ can reproduce the observed variability of stable water vapour isotopes on time scales of hours to days, and thus allows studying the mechanisms that control the subtropical free-tropospheric humidity. Changes of isotopic signals along backward trajectories from the Canary Islands region reveal the physical processes behind the short-term isotope variability. We identify four predominant moisture transport pathways of mid-tropospheric air, each with distinct isotopic signatures: (1) Air parcels originating from the convective boundary layer of the Saharan heat low (SHL). These are characterised by a homogenous isotopic composition with a particularly high $\delta D$ (median mid-tropospheric $\delta D = -122‰$), which results from dry convective mixing of low-level moisture of diverse origin advected into the SHL. (2) Air parcels originating from the free troposphere above the SHL. Although experiencing the largest changes in humidity and $\delta D$ during their subsidence over West Africa, these air parcels typically have lower $\delta D$ values (median $\delta D = -148‰$) than air parcels originating from the boundary layer of the SHL. (3) Air parcels originating from outside the SHL region, typically descending from tropical upper levels south of the SHL, which are often affected by moist convective injections from mesoscale convective systems in the Sahel. Their isotopic composition is much less enriched in heavy isotopes (median $\delta D = -175‰$) than those from the SHL region. (4) Air parcels subsiding from the upper-level extratropical North



Atlantic. This pathway leads to the driest and most depleted conditions (median $\delta D = -255‰$) in the middle troposphere near the Canary Islands. The alternation of these transport pathways explains to a large degree the observed high variability in humidity and $\delta D$ on synoptic time scales. We further show that the four different transport pathways are related to specific large scale-flow conditions. In particular, distinct differences in the location of the North African mid-level anticyclone and of

extratropical Rossby wave patterns occur between the four transport pathways. Overall, this study demonstrates that the adopted Lagrangian isotope perspective enhances our understanding of air mass transport and mixing and offers a sound interpretation of the free-tropospheric variability of specific humidity and isotope composition on time scales of hours to days in contrasting atmospheric conditions over the eastern subtropical North Atlantic.

## 1 Introduction

Understanding the subtropical atmospheric water cycle is of particular importance, since the free-tropospheric humidity and low-level cloud cover over the subtropical oceans strongly affect the global radiative balance via the greenhouse (Held and Soden, 2000; Schmidt et al., 2010) and albedo (Bony and Dufresne, 2005; Stephens, 2005) effects. The dryness of the free troposphere is primarily linked to the adiabatic descent of dehydrated air from the outflow of the Hadley circulation (Sun and Lindzen, 1993; Frankenberg et al., 2009) and the isentropic transport of very dry air by midlatitude eddies (Galewsky et al.,

2005; Cau et al., 2007). Moistening of the subtropical free-tropospheric air results from several processes including large-scale transport from the tropics (Pierrehumbert and Roca, 1998; Couhert et al., 2010; Knippertz et al., 2013), detrainment of condensate from convective clouds and its subsequent evaporation (Sun and Lindzen, 1993; Risi et al., 2008, 2010a), and vertical mixing associated with convection (Yang and Pierrehumbert, 1994; Lee et al., 2011; Brown et al., 2013). In addition, moisture export from the African continent over the subtropical North Atlantic has been observed in summer (González et al.,

2016; Lacour et al., 2017).

In summer, a near-surface thermal low-pressure system establishes over the Sahara, referred to as the Saharan heat low (SHL). The SHL is an important synoptic-scale weather system over West Africa (Lavaysse et al., 2010a, b) and notably a key element of the West African Monsoon system (Sultan and Janicot, 2003; Messager et al., 2010; Cornforth et al., 2017). Moreover, the SHL strongly influences the transport of air from north-western Africa over the adjacent subtropical North Atlantic (Lacour et

al., 2017). The low-level cyclonic circulation of the SHL strengthens the south-westerly monsoon flow and the north-easterly Harmattan flow, which results in an enhanced near-surface convergence along the so-called Intertropical Discontinuity. Dry convective mixing north of the Intertropical Discontinuity leads to the formation of a deep well-mixed boundary layer during the day and an anticyclonic circulation aloft (from about 700 hPa), which plays an essential role in the maintenance of the African easterly jet south of the SHL (Thorncroft and Blackburn, 1999), and in the transport of continental mid-tropospheric

air over the eastern subtropical North Atlantic. In this study, we aim to further disentangle the complex interplay between dehydrating and moistening processes that control the subtropical free-tropospheric moisture budget in the Canary Islands region, which is considered to be representative for the eastern subtropical North Atlantic. In particular, we seek to assess the



importance of the SHL dynamics for moistening the free troposphere. To this end we use multi-platform observations and regional simulations of stable water isotopes in atmospheric water vapour.

Stable water isotopes have proven to be highly useful to investigate the physical mechanisms involved in the atmospheric water cycle (Dansgaard, 1964; Gat, 1996; Galewsky et al., 2016). These natural tracers of water phase changes capture the moist diabatic history experienced by air parcels. Additionally, due to the distinct fingerprints of air masses with different origin, the isotopic composition of water vapour can provide information about atmospheric processes that do not involve phase changes, for instance, turbulent mixing or large-scale water vapour transport. The stable water isotope composition of a

water sample is usually quantified by the $\delta$ notation (Craig, 1961): $\delta(\text{D/H}) = \delta\text{D} = (R_{\text{sample}} / R_{\text{VSMOW}} - 1)$, where $R$ is the molecular ratio of the concentration of $\text{HD}^{16}\text{O}$ to the concentration of $\text{H}_2^{16}\text{O}$ and a stochastic isotope distribution across individual isotopologues is assumed. The $\delta$ notation expresses the relative deviation of $R$ from the internationally accepted primary water isotope standard, that is, the Vienna standard mean ocean water (VSMOW2; IAEA, 2017).

With recent technical advances in measuring water vapour isotope compositions in situ and with remote sensing, new

possibilities emerged for investigating governing processes of the atmospheric water cycle. Near-surface in situ measurements offer continuous records of the isotopic composition of the near-surface water vapour at high temporal resolution, thereby allowing, for instance, detailed analyses of surface evaporation processes (e.g., Aemisegger et al., 2014; Thurnherr et al., 2020). Airborne laser spectrometric measurements provide highly resolved in-situ profiles of the water vapour isotopic composition during field campaigns and are particularly beneficial to study small-scale processes such as cloud formation or

air mass mixing (e.g., Dyroff et al., 2015; Sodemann et al., 2017). In addition, ground-based (Schneider et al., 2012) and satellite-based (Worden et al., 2006; Frankenberg et al., 2009; Schneider and Hase, 2011; Lacour et al., 2012; Diekmann et al., 2021b) remote sensing systems enable observations of water vapour isotopes in the free troposphere on a quasi-global scale, which may document the influence of large-scale water vapour transport (e.g., Schneider et al., 2016; Lacour et al., 2017). As the various isotope measurement methods have complementary characteristic, e.g., in terms of temporal and spatial resolution,

a combination of several observational datasets provides an excellent opportunity to enhance our understanding of the mechanisms controlling tropospheric humidity.

Previous studies have demonstrated the potential of water vapour isotope observations to identify governing processes that affect the moisture budget in the subtropical troposphere such as evaporation from oceans (Steen-Larsen et al., 2014; Benetti et al., 2014; Bonne et al., 2019), local mixing between the marine boundary layer and the free troposphere (Noone et al., 2011;

Noone, 2012; Bailey et al., 2013; Benetti et al., 2015, 2018; Galewsky, 2018a, b), and large-scale dynamics (Galewsky and Hurley, 2010; Risi et al., 2010b; González et al., 2016; Schneider et al., 2016; Lacour et al., 2017; Aemisegger et al., 2020). Detrainment of moisture from the marine boundary layer to the subtropical free troposphere has been attributed to turbulent mixing and shallow convection (Bailey et al., 2013). But moisture detrainment from the marine boundary layer is strongly limited by the temperature inversion and primarily affects the moisture budget in the lower troposphere at the local scale

(Galewsky, 2018a, b). Therefore, mid-tropospheric moisture is mainly influenced by large-scale transport of different air masses that experience isentropic and cross-isentropic mixing (Galewsky et al., 2007; Noone et al., 2011; González et al.,



2016; Schneider et al., 2016; Lacour et al., 2017). Based on isotope observations, it could be shown that variations in the mid-tropospheric moisture and isotope composition over the eastern subtropical North Atlantic are linked to the alternating transport of dry, low-$\delta$D air from the upper-level extratropical North Atlantic and moist, high-$\delta$D air from North Africa (González et al., 2016; Schneider et al., 2016; Lacour et al., 2017). In addition, it was found that the transport from Africa has a strong seasonal cycle with a clear maximum in summer, which is closely related to the activity of the SHL (Lacour et al., 2017). Even though different components of the subtropical free-tropospheric water cycle could be identified so far with the help of isotope observations, the attribution of observed isotope signals to individual meteorological processes remains challenging. Due to the complex nature of the involved dynamical and physical processes, numerical models are essential for a more detailed interpretation of the isotope observations and to fully exploit their potential.

Stable water isotope physics has been implemented in several global (e.g., Risi et al., 2010c; Werner et al., 2011) and regional atmosphere circulation models (e.g., Pfahl et al., 2012). These Eulerian models include a detailed representation of relevant processes of the atmospheric moisture cycle and provide the full four-dimensional isotope fields. Isotope-enabled global circulation models have a relatively coarse spatial resolution and are thus suited for investigating long-term isotopic signals, e.g., in paleoclimate archives. For more detailed, process-related studies of synoptic-scale variability, isotope-enabled regional circulation models are better suited (e.g., Sturm et al., 2005; Blossey et al., 2010; Yoshimura et al., 2010; Pfahl et al., 2012). Pfahl et al. (2012) incorporated isotopes into the regional weather and climate prediction model COSMO (Steppeler et al., 2003; Baldauf et al., 2011) with an advanced microphysical scheme and nonhydrostatic dynamics. Several case studies have demonstrated that COSMO$_{iso}$ simulates isotope variability at high spatial and temporal resolution well and is thus suitable to investigate the governing mechanisms of the atmospheric water cycle ranging from small-scale microphysical processes to synoptic-scale weather (Pfahl et al., 2012; Aemisegger et al., 2015; Dütsch et al., 2016, 2018; Christner et al., 2018; Diekmann et al., 2021a).

In this study, we use dedicated high-resolution COSMO$_{iso}$ simulations in order to assess the dynamical and physical processes behind the hourly to daily variations of the mid-tropospheric isotope composition in the Canary Islands region, and to determine the connection of this isotope variability to moisture transport from different regions. The isotope simulation is complemented by kinematic backward trajectories computed from three-dimensional COSMO$_{iso}$ wind fields. Using trajectories allows us to assess whether the observed contrasting isotope signals are related to transport from the North Atlantic vs. North Africa and more specifically from the SHL. The combination of the isotope-enabled model COSMO$_{iso}$ with the Lagrangian diagnostics and the multi-platform water vapour isotope observations provides a solid framework to analyse and explain the observed atmospheric isotope signals. In turn, the high-resolution isotope observations allow a robust evaluation of physical processes in the model, which are difficult to constrain by measurements of specific humidity alone. This study aims to 1) validate a COSMO$_{iso}$ simulation with explicit moist convection against multi-platform observations of stable isotopes in water vapour, 2) identify predominant moisture transport pathways in the free troposphere in the Canary Islands region and investigate the associated synoptic-scale flow patterns and the physical processes that shape the observed isotope signals, and 3) quantify the climatological importance of the identified transport pathways. The paper is structured as follows: an overview of different



water vapour isotope observations used in this study as well as a description of the COSMO$_{iso}$ model is given in Section 2. The results are presented and discussed in Sections 3 (COSMO$_{iso}$ validation) and 4 (analysis of transport pathways). Finally, a concluding summary of the main results is provided in Section 5.

## 2 Data and Methods

### 2.1 Observations of water vapour isotopes

We use ground- and space-based remote sensing observations as well as airborne in situ measurements of tropospheric water vapour isotopes from the project MUSICA (MUlti-platform remote Sensing of Isotopologues for investigating the Cycle of Atmospheric water). This combination of multi-platform observations allows a comprehensive model validation, since the different isotope measurement methods are sensitive to different vertical, horizontal, and temporal scales. Dyroff et al. (2015)

and Schneider et al. (2016) gave a detailed overview of the campaign and the measurements.

### 2.1.1 Airborne in situ measurements

The airborne in situ measurements are performed with the ISOWAT II tuneable diode laser spectrometer (Dyroff et al., 2015) during the 7-day MUSICA aircraft campaign in July and August 2013. The instrument has been specifically designed for measuring specific humidity $q_v$ and $\delta$D in water vapour by means of laser absorption spectroscopy aboard research aircrafts.

Vertical profiles of $q_v$ and $\delta$D in water vapour are measured between sea level and around 7 km altitude in the Canary Islands region (Fig. 1). The temporal resolution of the measurements is 1 s, corresponding to a horizontal resolution of about 80 m and a vertical resolution of about 3 m. For the comparison with COSMO$_{iso}$, we average the ISOWAT measurements every minute, which results in a horizontal and vertical resolution of about 5 km and 180 m, respectively. The high temporal and spatial resolution of the laser spectroscopic measurements enables an accurate sampling of the isotopic composition in water

vapour and is thus suitable for the validation and investigation of small-scale processes. However, since the data is only available for a few flight days in a limited area, it is less appropriate to study large-scale water vapour transport pathways.

The uncertainty of the ISOWAT $\delta$D measurements depends on the absolute humidity and is around 10‰ for most conditions during the campaign. For very dry conditions, as encountered in the arid upper troposphere, the uncertainty is higher and can exceed 34‰ for water vapour mixing ratios below 500 ppmv (corresponding to a specific humidity of 0.31 g kg$^{-1}$). The

uncertainty estimates are based on calibration measurements performed before and after each flight and additionally validated by in-flight calibrations at varying altitudes. More details about the calibration methods and the campaign in general can be found in Dyroff et al. (2015).

In order to compare the COSMO$_{iso}$ simulation with the airborne in situ measurements, hourly COSMO$_{iso}$ fields of $q_v$ and $\delta$D in water vapour are temporally and spatially interpolated along the flight track. In addition, we sample minimum and maximum

values in a horizontal 5° x 5° box around each point along the flight tracks, which reflects the synoptic-scale variability (Fig.




1). This uncertainty measure accounts for the heterogeneous meteorological conditions and large gradients that often occurred during the flights (see appendix A), and indicates whether discrepancies between measurements and COSMO$_{iso}$ can be partially explained by small shifts of the simulated gradients.

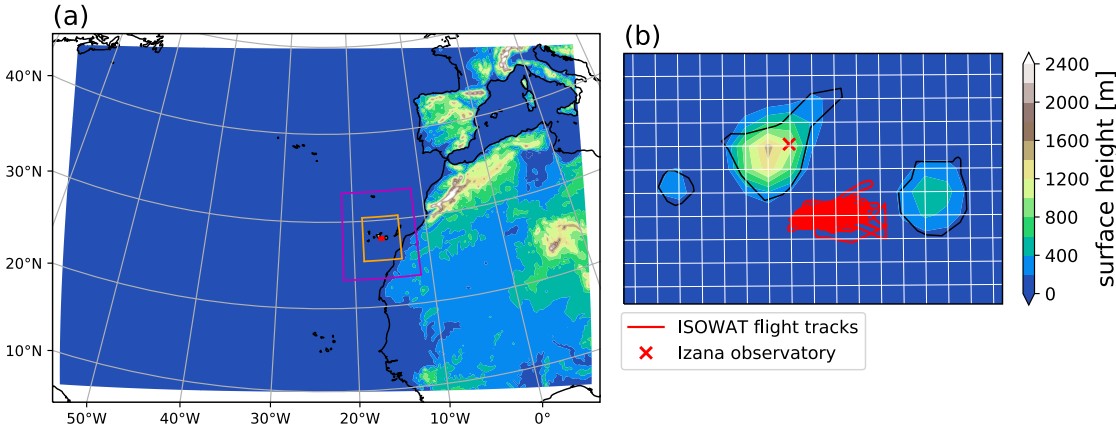

Figure 1: (a) COSMO$_{iso}$ model domain and topography. The orange box indicates the sampling area for the uncertainty estimation used for the comparison between COSMO$_{iso}$ and airborne in situ water vapour isotope data. The magenta box represents the sampling area of the satellite-based IASI remote sensing observations of stable isotopes in water vapour. (b) Enlarged view of the Canary Islands. The red lines depict the flight tracks of the summer 2013 MUSICA aircraft campaign and the red cross shows the location of the Izaña observatory on Tenerife, where the Fourier transform infrared (FTIR) spectrometer is installed for the ground-based remote sensing. The white lines represent the COSMO$_{iso}$ grid with a horizontal resolution of 0.125° (~ 14 km).

### 2.1.2 Ground-based remote sensing observations

Ground-based remote sensing observations can deliver continuous observations of the isotopic composition of the free-tropospheric water vapour at a specific location. The observational data used in this study is generated from the Fourier Transform InfraRed (FTIR) system located at the Izaña observatory (2370 m a.s.l.) on Tenerife (Fig. 1b), which is part of the Network for the Detection of Atmospheric Composition Change (NDACC). The FTIR spectrometer records high resolution solar absorption spectra allowing the retrieval of volume mixing ratios of $H_2^{16}O$, $H_2^{18}O$ and $HD^{16}O$ by using the NDACC MUSICA retrieval processor (Barthlott et al., 2017). The MUSICA NDACC/FTIR retrievals (for brevity denoted as FTIR retrievals in the following) are empirically validated against in situ measurements made by the aircraft ISOWAT instrument and by two commercial Picarro laser spectrometers installed at two different sites on Tenerife (Izaña, 2370 m a.s.l. and Teide, 3550 m a.s.l.; Schneider et al., 2015). This quality assessment guarantees a robust estimate of random and systematic errors, which for $q_v$ amount to 2% and 10%, respectively, and for $\delta D$ to 25‰ and 150‰. While random errors are dominated by uncertainties in the atmospheric temperature profiles and artefacts in the spectral baseline, systematic errors are attributed to uncertainties in the spectroscopic parameters. A comprehensive description of the MUSICA NDACC/FTIR remote sensing retrieval method and error estimation was presented in Schneider et al. (2012) and Barthlott et al. (2017).

Two different data types of FTIR observations are available (Barthlott et al., 2017). The first type, the so-called type 1 product, is the direct retrieval output and offers best estimates of $q_v$ in the lower, middle and upper troposphere. This data is used for





validating $q_v$ simulated in COSMO$_{iso}$. The second type, the type 2 product, is the a posteriori processed retrieval output. It reports the best estimation of $\{q_v, \delta D\}$ pairs and provides profiles of the isotopic composition of water vapour for the lower and middle troposphere. The a posteriori correction assures that the $q_v$ and $\delta D$ products represent the same atmospheric air

mass by adjusting the much finer vertical resolution of the derived $q_v$ profile to the vertical resolution of the $\delta D$ profile. In addition, the correction minimises cross-dependencies of retrieved $\delta D$ concentrations on actual atmospheric $q_v$ concentrations. We use this data product for the comparison of either $\delta D$ values or $\{q_v, \delta D\}$-pair distributions from COSMO$_{iso}$ with FTIR data. The remote sensing water vapour isotope concentrations are not representative for a single altitude but rather reflect the atmospheric situation averaged over a vertical layer. The smoothing of the real atmospheric profile by the remote sensing

measurement process is described by the averaging kernel. Figure 2 shows a typical averaging kernel of an a posteriori corrected $\delta D$ retrieval at 4.9 km a.s.l., which indicates that the retrieved $\delta D$ value at 4.9 km mostly reflects atmospheric $\delta D$ concentrations between 3 km and 7 km. The averaging kernel matrix $\boldsymbol{A}$ is an important output of the retrieval and specifies the response of the retrieved concentration profile $\hat{\boldsymbol{x}}$ to variations in the real atmospheric concentration profile $\boldsymbol{x}$:

$$\hat{\boldsymbol{x}} = \boldsymbol{A}\,(\boldsymbol{x} - \boldsymbol{x}_a) + \boldsymbol{x}_a,$$


where $\boldsymbol{x}_a$ is an a priori atmospheric concentration profile towards which the retrieval is constrained. The atmospheric concentration profiles are vectors with $3 \times 23$ entries specifying the three isotope species concentrations at 23 altitude levels between the surface altitude at the Izaña Observatory (2.37 km a.s.l.) and 55.3 km a.s.l. (what is used as top of the atmosphere in the FTIR retrieval).

In order to quantitatively compare COSMO$_{iso}$ data with FTIR observations, the hourly COSMO$_{iso}$ output needs to be processed such that it has the same characteristics as the remote sensing products. Convolving the modelled water vapour isotope concentration profile at a specific grid point $\boldsymbol{x}_{COSMO}$ with the averaging kernel matrix from the FTIR retrieval yields a water vapour isotope concentration profile $\hat{\boldsymbol{x}}_{COSMO}$ as would have been observed by FTIR system in the atmosphere simulated by COSMO$_{iso}$:

$$\hat{\boldsymbol{x}}_{COSMO} = \boldsymbol{A}\,(\boldsymbol{x}_{COSMO} - \boldsymbol{x}_a) + \boldsymbol{x}_a,$$


where $\hat{\boldsymbol{x}}_{COSMO}$, $\boldsymbol{x}_{COSMO}$, and $\boldsymbol{x}_a$ are vectors with the three isotope species concentrations specified at the 17 lowest FTIR levels. Vertical levels above the model top at 23.6 km are truncated.

In this study, we compare water vapour isotope concentrations retrieved at 4.9 km altitude, which corresponds to the FTIR

retrieval level with the highest sensitivity. Moreover, we consider only observations retrieved around midday with solar zenith angles smaller than 30°. Morning and evening retrievals have much broader averaging kernels, i.e., the vertical sensitivity is reduced. The small zenith angles of the considered retrievals further imply that the distance between the FTIR measurement site and the point of atmospheric observation at 4.9 km is much smaller than the horizontal grid spacing in COSMO$_{iso}$. We therefore take the COSMO$_{iso}$ water vapour isotope profile at the grid point closest to the FTIR station for the comparison with





the remote sensing observations (Fig. 1b). The eight neighbouring grid points are used to estimate the spatial variability of
COSMO$_{iso}$, which is defined by the minimum and maximum $\delta D$ and $q_v$ values of the postprocessed COSMO$_{iso}$ water vapour
isotope profiles. Finally, we only use FTIR retrievals associated with no clouds in the COSMO$_{iso}$ simulation at the
corresponding grid point (based on the cloud area fraction output from the model), because the remote sensing retrieval
processes only spectra measured for cloud-free conditions.

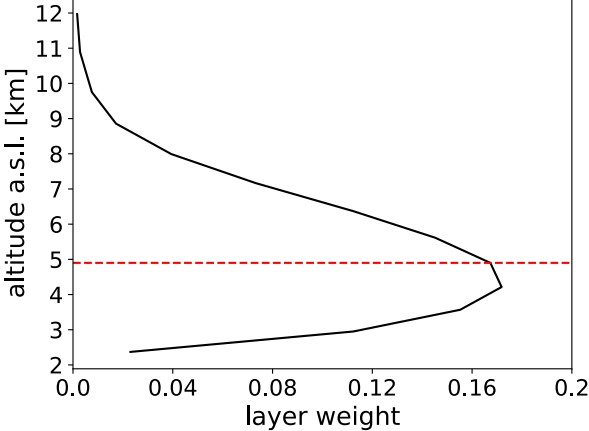


**Figure 2: Averaging kernel of the a posteriori corrected $\delta D$ proxy state retrieved at 12:06 UTC 26 July 2013 from the ground-based
Fourier transform infrared (FTIR) spectra at Izaña for the 4.9 km retrieval level (red line).**

### 2.1.3 Space-based remote sensing observations

In addition to the ground-based FTIR observations, isotope observations are retrieved by the MUSICA IASI processor using
the spectral radiances measured by the Infrared Atmospheric Sounding Interferometer (IASI) onboard the polar orbiting
satellites Metop-A and Metop-B. The IASI sensor measures the thermal infrared emission of the Earth's surface and the
emission and absorption in the atmosphere with a high horizontal resolution of 12 km at nadir with a global coverage twice
per day. The high-quality IASI spectra allow a simultaneous and combined optimal estimation of the volume mixing ratios of
$H_2^{16}O$ and $HD^{16}O$ as well as of the ratio product $\delta D$ in the free troposphere  by considering the MUSICA IASI retrieval
procedure of Schneider and Hase (2011) and including the retrieval updates as described in Schneider et al. (2021). Analogous
to the ground-based data, an additional post-processing step creates the optimal estimation $\{q_v, \delta D\}$-pair product (type 2
product, see Sect. 2.1.2).  This product has its maximum sensitivity at a height of approximately 4.2 km. The typical
uncertainties are around 5–10% for $q_v$ and 10–30‰ for $\delta D$ (Diekmann et al., 2021b). The MUSICA IASI retrieval product is
only available for cloud-free conditions.

Similar to the ground-based FTIR observations, an averaging kernel is applied to COSMO$_{iso}$ output for the comparison of
simulated data with IASI observations. The original IASI retrieval simulator by Schneider at al. (2017) uses a very simple
radiative transfer model. Recently, the retrieval simulator has been improved and now uses the full radiative transfer code from
the MUSICA IASI retrieval processor. Since the MUSICA IASI retrievals only consider cloud-free observations, we perform



a rough cloud filtering for COSMO$_{iso}$ based on the cloud diagnostics of the model, i.e., total cloud cover, cloud water $q_c$ and
cloud ice $q_i$. Afterwards, we multiply the COSMO$_{iso}$ water vapour isotope concentration profile with the simulated kernels,
create the $\{q_v, \delta D\}$-pair product and obtain a water vapour isotope concentration profile as would have been observed by IASI
in the atmosphere simulated by COSMO$_{iso}$. In its most recent version, the retrieval simulator is capable of reproducing 95%
of the variability as observed by IASI.

The processed COSMO$_{iso}$ water vapour isotope profiles are directly compared to the IASI observations. Specifically, we
evaluate COSMO$_{iso}$ isotope concentrations retrieved at 4.2 km altitude in a 10° x 10° box (corresponding to approximately $10^6$
km$^2$) centred around Tenerife against IASI observations from the same retrieval altitude and sampling region. The statistical
comparison of this large number of independent observations (18,063 IASI retrievals) enables a robust model validation and
in turn, a detailed interpretation of space-based remote sensing observations of stable isotopes in water vapour.

## 2.2 COSMO$_{iso}$

The limited-area model COSMO$_{iso}$ (Pfahl et al., 2012) is an isotope-enabled version of the non-hydrostatic numerical weather
and climate prediction model COSMO (Steppeler et al., 2003). The isotope implementation incorporates two parallel, purely
diagnostic water cycles for the heavy water isotopes H$_2^{18}$O and HD$^{16}$O, which experience exactly the same physical processes
as H$_2^{16}$O except for fractionation during phase changes. The prognostic isotope multilayer soil model TERRA$_{iso}$ couples the
isotope content of the soil to COSMO$_{iso}$ (Dütsch, 2016; Christner et al., 2018), thereby accounting for fractionation during soil
evaporation. Plant transpiration is treated as non-fractionating and transmits the isotope signals of the contributing soil layers
according to the vegetation's rooting depths.

The COSMO$_{iso}$ simulation used in this study extends from 1 July to 31 August 2013. This time period includes the summer
2013 MUSICA campaign (see Sect. 2.1). The model domain was chosen such that it covers large parts of the North Atlantic,
West Africa and parts of the Mediterranean (Fig. 1a). With this large domain we intend to encompass all relevant source
regions of moisture for the SHL, which can be convectively lifted into the Saharan air layer. The simulation is performed with
explicit convection at a horizontal grid spacing of 0.125° (in rotated coordinates, corresponding to approximately 14 km) and
with 60 hybrid levels in the vertical. We also performed a simulation with the same horizontal resolution and with
parameterised convection, which however led to larger model biases in comparison with airborne, ground- and space-based
observations (see appendix B). This result is consistent with Marsham et al. (2013b), Pearson et al. (2014), and Pante and
Knippertz (2019), who showed that explicit convection leads to a more realistic representation of the West African Monsoon
already with a relatively coarse model resolution on the order of 10 km. Initial and lateral boundary conditions are provided
every 6 hours by the isotope-enabled global climate model ECHAM5-wiso (Werner et al., 2011) at a spectral resolution of
T106 (corresponding to a horizontal grid spacing of approximately 1°) and on 31 vertical levels. In order to keep the long
simulation close to reality, horizontal winds in COSMO$_{iso}$ at 850 hPa and above are spectrally nudged towards ECHAM5-
wiso. The nudging is performed at every time step (60 s) and operates at zonal and meridional wavenumbers of 5 and less. All
other COSMO$_{iso}$ fields run freely in the model domain. The spectral nudging technique at small wavenumbers forces the large-



scale flow in the limited-area model towards the large-scale flow in the global climate model without directly affecting small-scale weather features (von Storch et al., 2000; Schubert-Frisius et al., 2017). The ECHAM5-wiso, in turn, was nudged towards the ERA-Interim reanalysis dataset from the European Centre for Medium Range Weather Forecasts (Dee et al., 2011). The
ECHAM5-wiso nudging also includes temperature and surface pressure.

**2.3 Identification of the Saharan Heat Low**

To identify the SHL location from the COSMO$_{iso}$ simulation output, we use the methodology proposed by Lavaysse et al. (2009). The heat low detection criterion relies on the cumulative probability distribution of low-level atmospheric thickness between 700 and 925 hPa at 06 UTC over West Africa. A high percentile of this variable is used as a threshold to define the
area of the SHL. Unlike Lavaysse et al. (2009), we always average the thickness values over two days before computing the cumulative probability distribution in order to have a smoother transition of the SHL location between two consecutive days. Moreover, in contrast to Lavaysse et al. (2009), we use the 80$^{th}$ percentile instead of the 90$^{th}$ percentile since the COSMO$_{iso}$ domain is smaller than the area used by Lavaysse et al. (2009), which extends about 20° further east and 5° further south. Comparing the identified SHLs in COSMO$_{iso}$ and ERA-Interim, the latter calculated from the 90$^{th}$ percentile in the larger
domain, confirms that our chosen threshold leads to a realistic identification in terms of location and spatial extent. The SHL masks used in this study only contain grid points over continental Africa, i.e., points over the adjacent subtropical North Atlantic are not considered. Finally, only SHL features with a circumference larger than 1000 km are considered in order to avoid a fragmented structure of the SHL, which could occur due to the high spatial resolution of COSMO$_{iso}$. The top of the SHL is defined by the planetary boundary layer height as diagnosed in COSMO$_{iso}$ with a bulk Richardson number criterion.

**2.4 Lagrangian methods**

For the interpretation of observed and simulated water vapour isotope signals, we use kinematic backward trajectories computed from three-dimensional hourly COSMO$_{iso}$ wind fields with the Lagrangian analysis tool LAGRANTO (Wernli and Davies, 1997; Sprenger and Wernli, 2015). The trajectories start every hour and from every 20 hPa between 900 and 300 hPa above Tenerife (16.48°W, 28.30°N). They are calculated 10 days backward in time or until they leave the model domain.
Different COSMO$_{iso}$ variables such as $\delta D$, $q_v$, $q_i$, $q_c$, rain water $q_r$, snow water $q_s$, surface precipitation, and boundary layer height are interpolated along each trajectory and stored together with the trajectory position every hour. In addition, we attribute each trajectory to one of four predominant transport pathways, which we define as follows for this study: 1) North African air originating from the convective boundary layer of the SHL (hereafter referred to as SHL BL), 2) North African air coming from the free troposphere above the SHL (SHL FT), 3) North African air from outside the SHL region (typically descending
from tropical upper levels south of the SHL; TRP AFR), and 4) North Atlantic air (typically subsiding from the upper-level extratropical North Atlantic; NA). We attribute a trajectory to the SHL BL regime if the air parcel was at least at one time step in the convective boundary layer of the SHL during its travel; to the SHL FT regime if the air parcel was at least at one time step above but never in the boundary layer of the SHL during its travel over the SHL; to the TRP AFR regime if the air parcel





was at least once above continental Africa but never in or above the SHL; and to the NA transport pathway if the air parcel
never was above continental Africa. We will investigate whether these four transport pathways have a distinct isotopic
signature and lead to contrasting atmospheric conditions on time scales of hours to days in the Canary Islands region.

We validate the different transport pathways in COSMO$_{iso}$ by comparing trajectories calculated with winds from COSMO$_{iso}$
and 6-hourly ERA-Interim fields. The ERA-Interim trajectories start every hour from the same starting positions as the
COSMO$_{iso}$ trajectories and run 10 days backward in time. The validation criterion is based on comparing the origin of the
COSMO$_{iso}$ trajectories with the origin of the ERA-Interim trajectories, which are regarded as "truth". We distinguish between
trajectories from continental Africa, and trajectories from the North Atlantic. For each arrival time and vertical level of the
trajectory starting profile, trajectories arriving within a 6-hour interval centred at the considered arrival time are compared. A
6-hour time window for the comparison is chosen to achieve a more robust validation. If at least four of the seven compared
trajectories in the considered time window agree on the origin (continental Africa vs. North Atlantic), the transport in
COSMO$_{iso}$ is considered "reliable".

## 3 Comparison of the COSMO$_{iso}$ simulation with multi-platform isotope observations

In this section, we compare the two-month COSMO$_{iso}$ simulation of July and August 2013 (Sect. 2.2) to multi-platform water
vapour isotope observations (Sect. 2.1) in order to quantitatively evaluate the performance of COSMO$_{iso}$ in modelling the free-
tropospheric variability of humidity and isotopic composition on time scales of hours to days in contrasting atmospheric
conditions over the eastern subtropical North Atlantic. To this end, we first compare COSMO$_{iso}$ $q_v$ and $\delta D$ in water vapour
against airborne in situ observations (Sect. 2.1.1). Given its high temporal and spatial resolution, this dataset is beneficial to
accurately evaluate the vertical distribution of water vapour isotopes in COSMO$_{iso}$. The availability of the data, however, is
restricted to a few flights. Therefore, we further compare mid-tropospheric $q_v$ and $\delta D$ signals in COSMO$_{iso}$ with ground- and
space-based remote sensing observations (Sect. 2.1.2 and 2.1.3) over the entire two-month period. The continuous availability
of remote sensing observations enables a statistically robust validation of the free-tropospheric isotope composition in
COSMO$_{iso}$.

### 3.1 Airborne in situ measurements

Figure 3 shows the comparison of $q_v$ and $\delta D$ in water vapour from COSMO$_{iso}$ with the aircraft observations. The flights were
characterised by a fast ascent up to 7 km altitude followed by a slow step-wise descent. They covered an area of about 50 km
in the zonal and 25 km in the meridional direction over the ocean south of Tenerife (Fig. 1b). Each of the seven flights was
performed in the morning during 1 to 2.5 hours, yielding a total flight time of about 14 hours. The vertical airborne profiles
reveal large variability in humidity and isotopic composition. Specifically, the measured $q_v$ and $\delta D$ values range from 0.08 g
kg$^{-1}$ to 17 g kg$^{-1}$ and from –590‰ to –50‰ along the flight paths with fast variations. COSMO$_{iso}$ reasonably captures this
observed short-term variability. While modelled and observed $q_v$ and $\delta D$ values agree well in the lower troposphere (mean





absolute differences at 0–2 km a.s.l.: $\Delta\ln(q_v)$ (0–2 km) = 0.35 g kg$^{-1}$, $\Delta\delta D$ (0–2 km) = 38‰), differences occur for both variables

in the middle and upper troposphere ($\Delta\ln(q_v)$ (2–7 km) = 0.64 g kg$^{-1}$, $\Delta\delta D$ (2–7 km) = 71‰) and are largest around 6 km a.s.l.

($\Delta\ln(q_v)$ (> 6 km) = 0.91 g kg$^{-1}$, $\Delta\delta D$ (> 6 km) = 114‰). At this altitude, $q_v$ and $\delta D$ tend to be too high in COSMO$_{iso}$ compared

to the measurements. These differences can be partly explained by synoptic-scale uncertainties in COSMO$_{iso}$ associated with

the very heterogeneous meteorological conditions that occurred during the flights (see appendix A for more details).

Considering the synoptic-scale variability of $q_v$ and $\delta D$ in COSMO$_{iso}$ in the Canary Islands region by sampling minimum and

maximum values in a horizontal 5° x 5° box around each data point along the flight path (red and blue shading in Fig. 3)

reveals that the differences are mostly not significant, except for 30 July. On this day, aircraft measurements show very strong

spatial gradients in $q_v$ and $\delta D$ around 5.5 km altitude, which are not represented in COSMO$_{iso}$ in the considered box. In general,

however, COSMO$_{iso}$ is able to capture the observed vertical humidity and isotope profiles in the middle to upper troposphere

albeit with some horizontal displacements.

The strong horizontal gradients in $q_v$ and $\delta D$ at 500 hPa (Figs. 4a,b) highlight the very heterogenous conditions in the Canary

Islands region (black box in Fig. 4) at 12 UTC 21 July 2013, which is representative for all flight days apart from 30 July.

Slight spatial shifts of these gradients can lead to a rather different isotope and humidity signal in the area of the flight tracks

(red box in Fig. 4). The comparison of $q_v$ at 500 hPa between COSMO$_{iso}$ and ERA-Interim (Fig. 4c) confirms that COSMO$_{iso}$

overall correctly simulates the large-scale distribution of the free-tropospheric moisture. Still, we can observe a north-westward

shift of the frontal zone towards the Canaries in COSMO$_{iso}$ compared to ERA-Interim, where the front is located farther to the

south-east along the African west coast. Due to the low spatial resolution of ERA-Interim, however, it is not clear how

accurately ERA-Interim represents this frontal situation. Figures 4e,f reveal why the quantified synoptic uncertainty in

COSMO$_{iso}$ does not reduce the differences between the modelled and observed $q_v$ and $\delta D$ values on 30 July 2013. On this day,

COSMO$_{iso}$ is far too moist in the Canary Islands region compared to ERA-Interim, and consequently also too enriched in heavy

isotopes (In the following, for reasons of readability, we will only use the term "enriched" when referring to "enriched in heavy

isotopes" and analogously for "depleted".). Since the large-scale atmospheric flow situation in COSMO$_{iso}$ agrees well with

ERA-Interim (see appendix A for more details), too excessive local convective mixing in the model is presumably responsible

for the observed humidity and isotope differences in the middle to upper troposphere. Previous studies have already reported

positive mid-tropospheric $\delta D$ biases in models due to an overestimated vertical moisture transport (Werner et al., 2011; Risi

et al., 2012; Christner et al., 2018). In addition, we have to consider that the very sharp vertical humidity gradients measured

by the ISOWAT spectrometer (Fig. 9 in Dyroff et al., 2015) cannot be fully resolved by COSMO$_{iso}$ with a vertical resolution

of about 400 m in the middle troposphere. The lower spatial and temporal resolution of the COSMO$_{iso}$ output fields compared

to the airborne in situ measurements constitutes in general a source of uncertainty for the validation of modelled $q_v$ and $\delta D$.







**Figure 3: Comparison of COSMO$_{iso}$ and airborne in situ measurements (ISOWAT) of $q_v$ and $\delta$D in water vapour from the MUSICA campaign in July and August 2013. The blue and red shadings represent the synoptic-scale variability of $q_v$ and $\delta$D in COSMO$_{iso}$ by quantifying the minimum and maximum values sampled in a 5° x 5° box around each data point on the flight path (see Sect. 2.1.1). The orange shading indicates the ISOWAT measurement uncertainty of $\delta$D.**

Figure 5a displays the aircraft observations and the corresponding COSMO$_{iso}$ data in $\{q_v, \delta$D$\}$ space. Both distributions have a very similar slope but the COSMO$_{iso}$ distribution covers a much narrower $\delta$D range and tends to be too enriched compared to the airborne data. Positive $\delta$D differences are present throughout the atmospheric column but most pronounced at low $q_v$ and $\delta$D values (Figs. 5a,b). The $q_v$ range covered by COSMO$_{iso}$ is comparable to the aircraft data and there is no clear tendency towards positive or negative differences from aircraft observations, except for the humid bias in COSMO$_{iso}$ at low $q_v$ (Fig. 5c). The representativeness of this model-aircraft comparison, however, is limited, since the data is not fully independent but comprise correlated observations from vertical $q_v$ and $\delta$D profiles that were performed at seven specific campaign days.





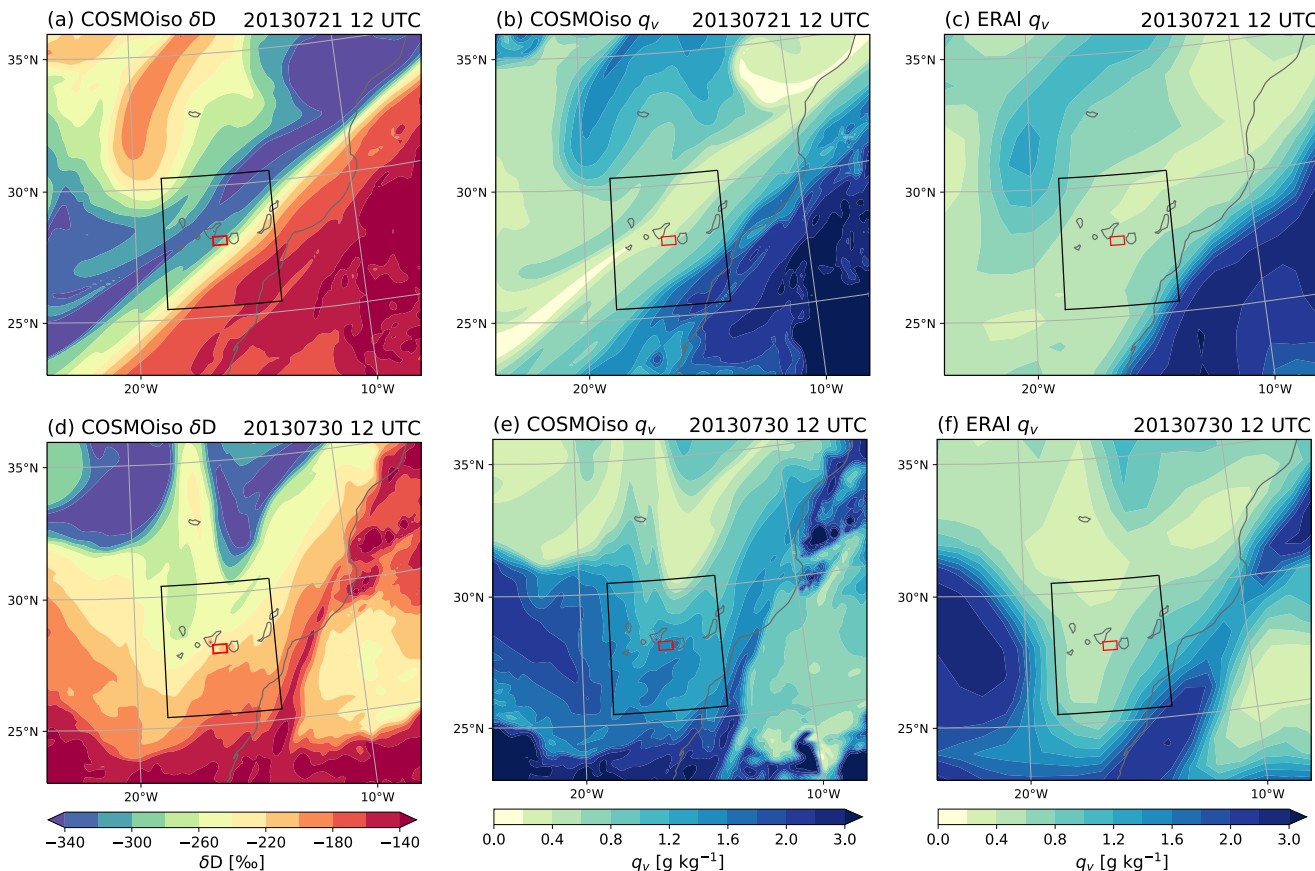

**Figure 4: COSMO$_{iso}$ and ERA-Interim moisture fields at 500 hPa at 12 UTC 21 July (a-c) and at 12 UTC 30 July (d-f) 2013. Panels (a,d) show COSMO$_{iso}$ $\delta$D in water vapour, (b,e) COSMO$_{iso}$ $q_v$, and (c,f) ERA-Interim $q_v$. The red box represents the flight area during the aircraft campaign, the black box the sampling area to estimate the COSMO$_{iso}$ synoptic-scale variability (see Sect. 2.1.1).**





**Figure 5: Comparison of COSMO_iso $q_v$ and $\delta$D in water vapour with multi-platform observations over the subtropical North Atlantic near Tenerife. (a-c) comparison with airborne in situ ISOWAT measurements performed between sea level and 7 km altitude during 7 flight days in July and August 2013, (d-f) comparison with satellite-based IASI remote sensing observations retrieved for 4.2 km in a 10° x 10° box centred around Tenerife during 32 days July and August 2013, (g-i) comparison with ground-based FTIR remote sensing observations retrieved for 4.9 km during 25 days in July and August 2013. Solid/dashed contours in panels (d-f) show the 50/90% frequency isolines of normalised two-dimensional histograms. Empty circles in panels (g-i) indicate a less reliable air parcel transport (see Sect. 2.4). The black box in panel (b) represents the $\delta$D space of panels (e,h).**



## 3.2 Space-based and ground-based remote sensing observations

In this section, we first compare satellite-based IASI observations with COSMO$_{iso}$ data that are adjusted according to the vertical characteristics of the IASI water vapour isotope product (Sect. 2.1.3). The datasets consist of 18,063 IASI and 196,744 COSMO$_{iso}$ samples that were retrieved at 4.2 km altitude in a 10° x 10° box centred around Tenerife (Fig. 1a) in July and August 2013. This large number of observational and model data enables a statistically robust model validation. Figure 5d shows the $\{q_v, \delta D\}$-pair distributions of COSMO$_{iso}$ and IASI. The distributions overlap to a great extent and cover ranges of $q_v$ from 0.4 g kg$^{-1}$ to 5 g kg$^{-1}$ and of $\delta D$ from –360‰ to –130‰, respectively. The slope of the COSMO$_{iso}$ $\{q_v, \delta D\}$ distribution is somewhat less steep than the slope of the IASI distribution, yielding a positive $q_v$ model bias for high $\delta D$ values ($\delta D > –180‰$) and a negative $q_v$ model bias for low $\delta D$ values ($\delta D < –320‰$) compared to IASI. Comparing the COSMO$_{iso}$ and IASI $\delta D$ distributions confirms the high consistency between the model and observations (Fig. 5e). Only a slightly negative model bias can be observed for higher $\delta D$ values ($\delta D > –260‰$). Similarly, the comparison of the COSMO$_{iso}$ and IASI $q_v$ distributions underlines the good agreement in the middle troposphere despite of small differences in the range of the highest and lowest $q_v$ values. Attributing the source of the $q_v$ and $\delta D$ differences in COSMO$_{iso}$ is difficult owing to the complex characteristic of the remote sensing retrievals and the post-processed COSMO$_{iso}$ data. Synoptic-scale uncertainties in the model, as discussed in the context of the aircraft observations in Sect. 3.1, constitute a minor source of the observed $q_v$ and $\delta D$ differences due to the large data sampling area.

The $\{q_v, \delta D\}$-pair distributions of the ground-based FTIR remote sensing observations (Fig. 5g; type 2 product; see Sect. 2.1.2) and the corresponding COSMO$_{iso}$ data multiplied with the FTIR kernels (see Sect. 2.1.2) are similar to the distributions shown for the COSMO$_{iso}$-IASI comparison (Fig. 5d). They cover a similar $\{q_v, \delta D\}$ range and in addition, the COSMO$_{iso}$ distribution also shows a slightly tilted slope compared to FTIR towards higher $q_v$ values for high-$\delta D$ air and lower $q_v$ values for low-$\delta D$ air. Likewise, the comparison of the FTIR (type 2 product) and COSMO$_{iso}$ $\delta D$ distributions indicates a tendency towards too low $\delta D$ values in the model compared to the observations (Fig. 5h). These differences, however, preliminary occur on days with a less reliable air parcel transport in the model (see Sect. 2.4). Uncertainties in the transport of air parcels mainly occur during transitions from the North Atlantic flow regime to an African flow regime and vice versa and are related to heterogeneous atmospheric conditions (as discussed in Sect. 3.1), which imply that subtle changes in the initialization of the trajectories (time, pressure) can lead to a completely different transport pathway. Finally, comparing modelled and observed (type 1 product; see Sect. 2.1.2) $q_v$ does not show a clear shift in COSMO$_{iso}$ (Fig. 5i), but in contrast to the comparison against space-based IASI data, the statistical representativeness of the ground-based remote sensing observations is limited. The 383 FTIR observations contain only a few independent samples since there are multiple observations per day in general (varying between 1 and 25 data points). Still, the high consistency between the comparisons with two different remote sensing datasets is encouraging and forms a sound basis for evaluating variations of $q_v$ and $\delta D$ in COSMO$_{iso}$ vs. FTIR observations on short time scales.





Figure 6 shows the time series of the COSMO$_{iso}$ and FTIR $q_v$ (type 1 product) and $\delta D$ (type 2 product) data. There is a pronounced day-to-day variability with $q_v$ ranging from 0.45 g kg$^{-1}$ to 4.3 g kg$^{-1}$ and $\delta D$ from –318‰ to –128‰. Conditions that are moist and enriched in heavy isotopes alter with dry and depleted conditions on a remarkably short time scale (e.g.,
from 18 to 19 July, or from 29 to 30 July). Comparing COSMO$_{iso}$ to the FTIR data demonstrates that our simulation accurately reproduces these observed temporal variations in mid-tropospheric $q_v$ and $\delta D$ despite occasional deviations. These differences can occur due to various factors. In the first place, we observe that the deviations are largest on days with a less reliable air parcel transport (discussed below in the context of Fig. 7). Synoptic-scale uncertainties in COSMO$_{iso}$, as already mentioned in Sect. 3.1, may thus partly explain the observed differences. Moreover, model uncertainties in the representation of complex,
sub-grid scale processes in the vicinity of the mountain slopes of Tenerife could also contribute to differences between the model and observations. Finally, there are also uncertainties associated with the remote sensing retrievals and the COSMO$_{iso}$ post-processing. Overall, however, we can conclude that COSMO$_{iso}$ realistically represents the variability in mid-tropospheric $q_v$ and $\delta D$ including fast day-to-day variations.

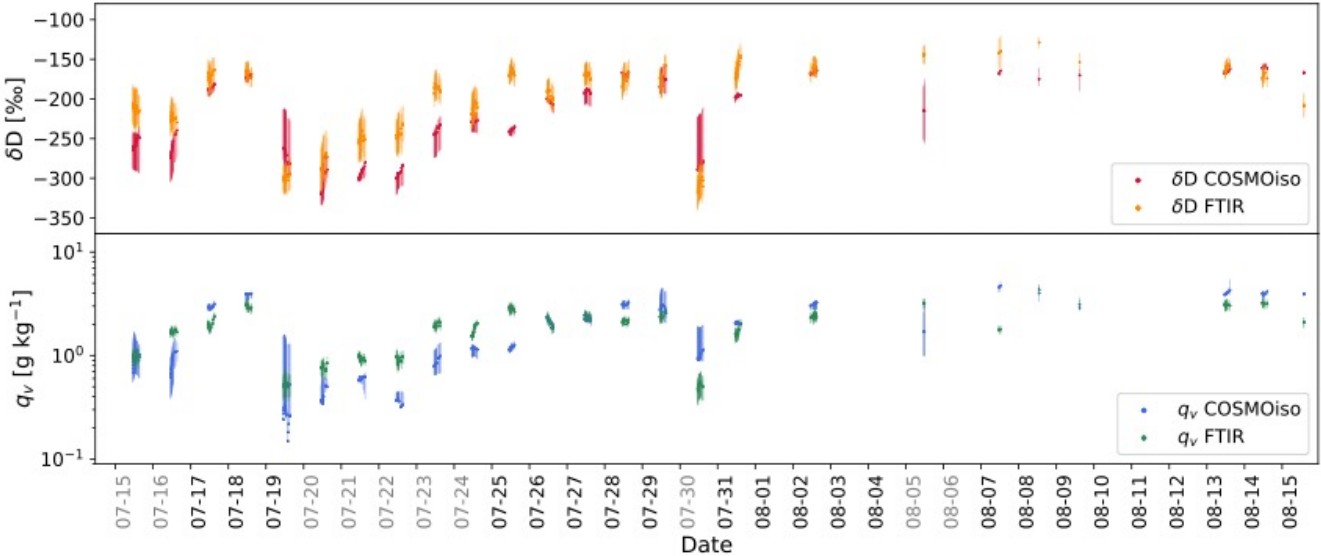

**Figure 6: Time series of COSMO$_{iso}$ and ground-based FTIR remote sensing $q_v$ and $\delta D$ obtained for the 4.9 km retrieval level (representative of 3–7 km) above Tenerife around midday. The shadings represent the estimated COSMO$_{iso}$ uncertainties (based on the minimum and maximum $\delta D$ and $q_v$ values of the eight neighbouring grid points) and the FTIR random errors. Gray dates on the x-axis indicate days with a less reliable air parcel transport (see Sect. 2.4).**

**4. Free-tropospheric moisture transport pathways over the eastern subtropical North Atlantic**

In this part, we investigate the governing processes behind the free-tropospheric isotope variability in the Canary Islands region observed in Sect. 3. First, the influence of different transport pathways on the free tropospheric humidity and isotopic variability in summer 2013 is evaluated. Secondly, the processes that occur along the four predominant transport pathways are investigated based on a detailed analysis of isotope signals along COSMO$_{iso}$ trajectories. Subsequently, large-scale circulation





anomalies in the mid troposphere associated with the four contrasting transport pathways are analysed. Finally, their relevance
is assessed climatologically.

## 4.1 Linking contrasting isotope conditions to different transport pathways

In Fig. 7, we present the $q_v$ and $\delta D$ composition in the vertical column between 300 and 900 hPa above Tenerife as simulated by COSMO$_{iso}$ for the period of July–August 2013. Substantial variability appears in the $q_v$ and $\delta D$ signals (Figs. 7a,b). In the middle troposphere, short intervals (1–5 days) of dry, low-$\delta D$ air ($\delta D < -260‰$) alternate with longer periods (3–11 days) of
humid, high-$\delta D$ air ($\delta D > -220‰$). Transitions between episodes often occur within a few hours and are characterised by sharp $q_v$ and $\delta D$ gradients (e.g., on 19 July and 17 August 2013; see also Figs. 4a–c). The enriched periods themselves also show fast, albeit less pronounced, variations in $\delta D$ on time scales of hours to days. These contrasting humidity and isotope conditions are closely linked to the four predominant transport pathways (see Sect. 2.4). While the dry and low-$\delta D$ air primarily comes from the North Atlantic (NA, blue shading in Fig. 7c), the humid and high-$\delta D$ air mainly originates from West Africa, where
we distinguish between air originating from the convective boundary layer of the SHL (SHL BL, red shading in Fig. 7c), the free troposphere above the SHL (SHL FT, purple shading in Fig. 7c), and tropical Africa south of the SHL (TRP AFR, green shading in Fig. 7c). Most pronounced temporal variations in $\delta D$ occur during transitions from an African transport regime to the NA transport regime and vice versa. For example, on 19 July and 17 August a strong depletion in heavy isotopes in mid-tropospheric water vapour from about $-160‰$ to $-320‰$ occurs within a few hours, simultaneously with a transition from an
African to the NA transport regime. The fast $\delta D$ variations during enriched periods can be partly explained by changes in the African transport pathways, where higher $\delta D$ signals tend to appear during the SHL BL transport regime compared to lower $\delta D$ signals during the TRP AFR transport regime (e.g., 20–23 August). Not surprisingly, air parcels from tropical Africa (TRP AFR) are more often affected by surface precipitation (black dots in Fig. 7c) and hydrometeors (not shown) than air parcels from the SHL region (SHL BL/FT). The formation of cloud water and precipitation directly influences the water vapour
isotopic composition of an air parcel, since heavy isotopes preferentially condense, whereby the remaining ambient water vapour becomes depleted in heavy isotopes. As a result, air parcels from TRP AFR are associated with lower $\delta D$ values due to condensation in moist convective systems in the Sahel compared to air parcels from the SHL region, where dry convection dominates. For the further analysis of physical and dynamical processes associated with the four contrasting transport pathways, we consider the air layer between 500 and 700 hPa (indicated by the black horizontal lines in Fig. 7). This layer is
particularly interesting from a dynamical viewpoint as the large-scale circulation dominates the humidity and isotope composition at these altitudes, whereas local mixing processes between the boundary layer and the free troposphere are less relevant than at lower levels. In addition, the remote sensing observations, against which we validated COSMO$_{iso}$, have their highest sensitivity around 600 hPa.

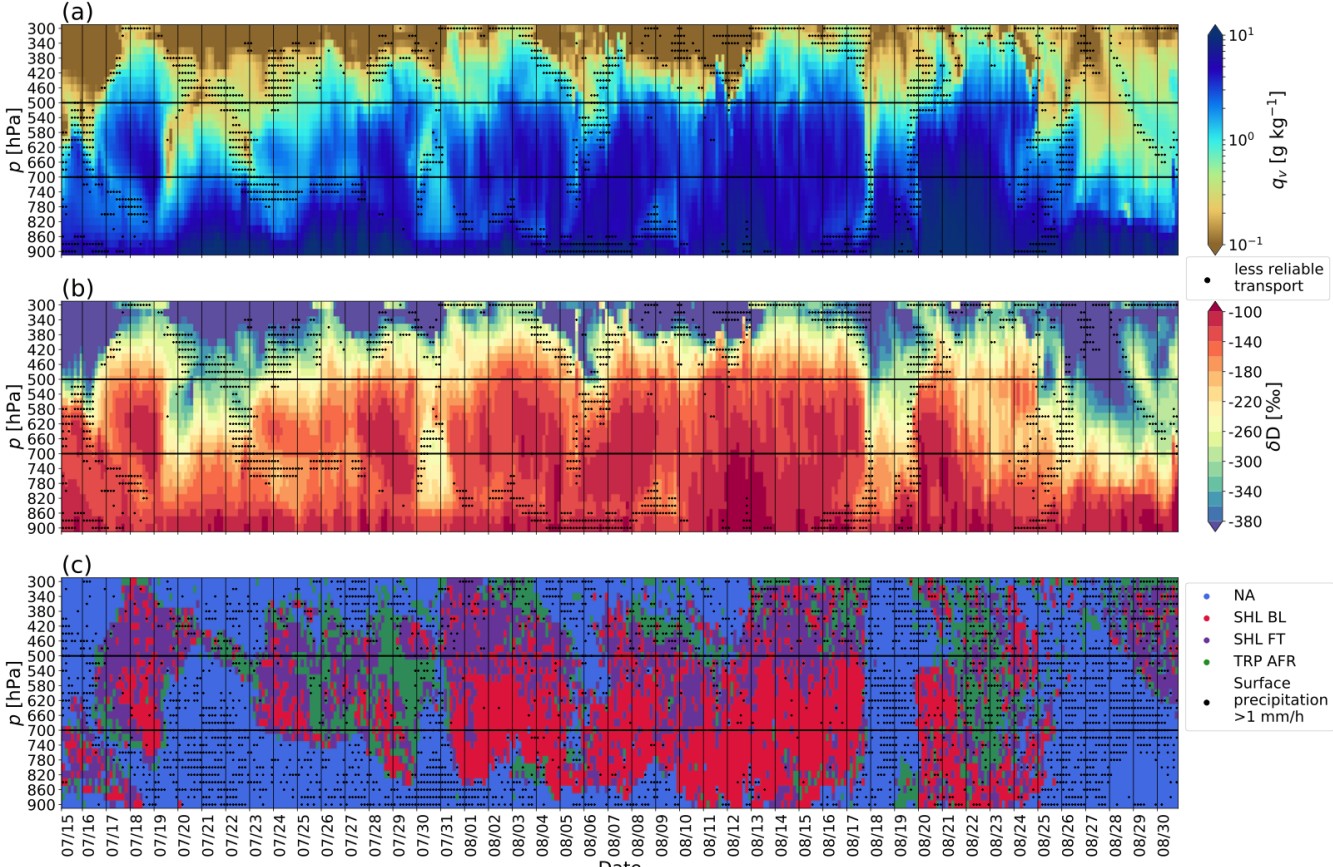

Figure 7: COSMO$_{iso}$ (a) $q_v$ and (b) $\delta D$ in water vapour 900–300 hPa above Tenerife, as well as (c) the Lagrangian origin of the air parcels. The black horizontal lines confine the horizontal air layer between 500–700 hPa used for further analyses in this study. Black dots in (a,b) indicate a less reliable air parcel transport (see Sect. 2.4). Black dots in (c) represent transport pathways with surface precipitation equal or larger than 1 mm h$^{-1}$ at least once over continental Africa for the African air parcels and over the North Atlantic for the North Atlantic air parcels, respectively.

The statistical analysis of the mid-tropospheric $\delta D$ signal highlights that each of the four transport pathways has a distinct isotopic signature (Fig. 8). The NA transport pathway is the most depleted (median $\delta D$ = –255‰) and most variable (interquartile range IQR = 92‰) regime, whereas the SHL BL pathway represents the most enriched (median $\delta D$ = –122‰) and most homogeneous (IQR = 21‰) category (Fig. 8a). Air parcels of the SHL FT (median $\delta D$ = –148‰, IQR = 47‰) and TRP AFR (median $\delta D$ = –175‰, IQR = 55‰) transport regimes are less enriched in heavy isotopes than air parcels of the SHL BL regime but more enriched than NA (Fig. 8a). In general, the three African transport pathways distinctly differ from the NA pathway with respect to $\delta D$ as IQRs of the distributions do not overlap (Fig. 8a). Likewise, we can observe clear differences between the {$q_v$, $\delta D$}-pair distributions of the African and NA transport pathways (Fig. 8b). While the $q_v$ ranges of the SHL FT and TRP AFR distributions overlap to some extent with the NA distribution, the $\delta D$ range does not for the 60%





contour. This distinct isotopic separation of the different transport pathways emphasizes the added value of water vapour
isotopes for investigating physical processes and transport pathways that affect tropospheric humidity. Figure 8b further
highlights the homogeneous character of the SHL BL category as well as its particularly high $\delta$D and $q_v$ values. By contrast,
the $\{q_v, \delta D\}$-pair distribution of the NA category reflects the dryness and depletion in heavy isotopes of this transport pathway.

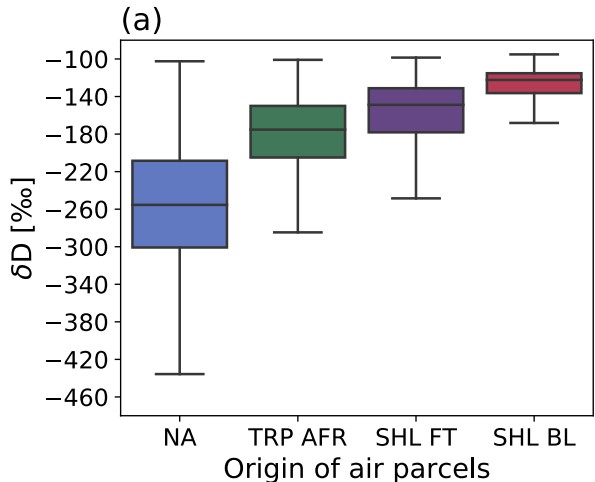

**Figure 8: Isotopic signature of COSMO$_{iso}$ air parcels arriving at 500–700 hPa above Tenerife and originating from the upper-level extratropical North Atlantic (NA, 26% occurrence frequency), from tropical Africa (TRP AFR, 16%), from the upper levels above the Saharan heat low (SHL FT, 26%) and from the Saharan heat low (SHL BL, 32%). (a) Boxplots of the $\delta$D signal separated by air parcel origin. The boxplots show the interquartile range by the extent of the box and the median by the black line in the box. The whiskers correspond to 1.5 times the proportion of the interquartile range past the lower and upper quartiles. (b) Relation between $\delta$D in water vapour and $q_v$. Solid/dashed contours show 30/60% frequency isolines of normalised two-dimensional histograms of respective transport regimes.**

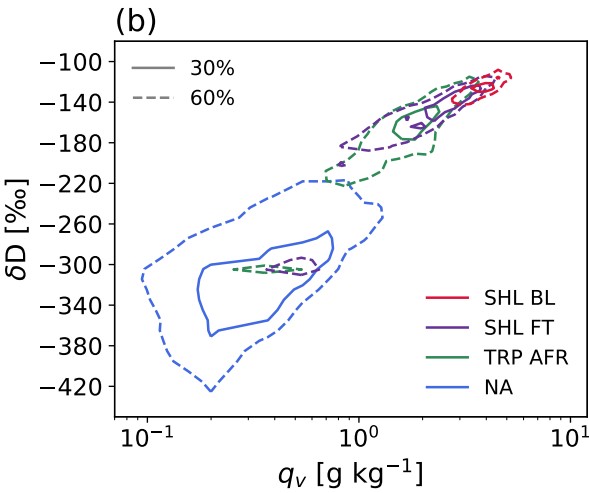

## 4.2 Characterisation of predominant transport pathways in the Canary Islands region

Detailed analysis of isotopic signals along backward trajectories provides valuable insights into the process history of air
parcels. Figure 9 summarises the median 10-day Lagrangian evolution of pressure $p$, $\delta$D and $q_v$ of the four different transport
pathways identified in this study. The median SHL BL transport pathway involves weak subsidence from about 590 to 620
hPa ($\Delta p_{5\text{-}10d}$ = 30 hPa (5 d)$^{-1}$) 5–10 days before arrival, followed by a lifting to 590 hPa ($\Delta p_{2\text{-}5d}$ = −30 hPa (3 d)$^{-1}$) 2–5 days
prior to arrival and a subsequent descent to 610 hPa ($\Delta p_{2d}$ = 20 hPa (2 d)$^{-1}$) during the last two days before arrival over the



Canary Islands. The initial subsidence goes along with a slight increase of $\delta D$ ($\Delta\delta D_{5-10d} = 32‰$ (5 d)$^{-1}$) and $q_v$ ($\Delta q_{v,5-10d} = 0.5$ g kg$^{-1}$ (5 d)$^{-1}$). The subsequent lifting occurs without changes in the median $\delta D$ and $q_v$. The SHL BL transport pathway is

further characterised by substantial variability in $p$ and $\delta D$ 5–10 days before arrival, which considerably reduces in the last four days before arrival. Both are related to SHL dynamics. Convective mixing in the SHL of dry, low-$\delta D$ air from the upper-level extratropics with moist, high-$\delta D$ air from low levels effectively homogenises the isotopic composition, producing a well-mixed and relatively homogeneous air mass subsequently travelling towards the Canary Islands. The homogeneous, enriched composition of the SHL BL air is a unique characteristic of this transport pathway (Fig. 8a) and a clear imprint of the dry

convective mixing in the SHL. Figures 10a,b show a representative example of the SHL BL transport pathway. Air parcels with a low $\delta D$ descend from the upper-level extratropical North Atlantic around the Atlas Mountains into the SHL (red squares in Fig. 10a), where they experience considerable enrichment in heavy isotopes and moistening due to dry convective mixing with high-$\delta D$ air from lower levels. The moisture of these low-level air parcels presumably originates from evaporation over the eastern Mediterranean and the Black Sea, as identified with extended ERA-Interim backward trajectories (Fig. 11). Two

principal moisture transport pathways dominate the low-level SHL inflow at the time when the air parcels in Fig. 10a,b are located in the SHL (Fig. 11). The first pathway, the so-called south-westerly monsoon flow, brings very humid air from the tropical Atlantic into the southern part of the SHL. The second pathway, the so-called north-easterly Harmattan flow, transports air from north-eastern Europe into the central and northern part of the SHL. These low-level air parcels experience a considerable moisture uptake during their transport across the eastern Mediterranean, with minor contributions from the Black

Sea, before they mix with upper-level extratropical air in the SHL. The well-mixed, isotopically homogeneous air then rapidly moves as a coherent air mass over the adjacent North Atlantic directly towards Tenerife (Figs. 10a,b). This direct and fast transport of air towards the Canary Islands is typical for the SHL BL transport pathway. It essentially conserves the isotopic imprint of the dry convective mixing in the SHL, which is reflected by the constant $\delta D$ signal along the trajectories during the last four days before arrival (Fig. 9).

Although originating from a similar region, air parcels of the SHL FT category follow a different pathway (Fig. 9). Typically, air parcels with a very low initial $\delta D$ (median $\delta D = -370‰$) from the upper troposphere experience a dramatic median increase in $\delta D$ ($\Delta\delta D_{10d} = 213‰$ (10 d)$^{-1}$) and a strong moistening ($\Delta q_{v,10d} = 2.1$ g kg$^{-1}$ (10 d)$^{-1}$) along with a strong subsidence ($\Delta p_{10d} = 220$ hPa (10 d)$^{-1}$) on their 10-day travel towards the Canary Islands. This indicates that at least in our simulation, there is vigorous exchange across the inversion on top of the SHL. This transport pathway exhibits the largest median changes in

humidity and isotopic composition among the four identified transport pathways. The pronounced enrichment in heavy isotopes usually occurs in the free troposphere above the SHL due to mixing in of high-$\delta D$ air that is injected from the boundary layer of the SHL into the free troposphere. Figures 10c,d illustrate how upper-level air parcels get enriched while anticyclonically rotating above the SHL (purple squares in Fig. 10c) before they move off the African coast and slowly approach the Canary Islands from the south. Note that these trajectories move slower than the ones in the SHL BL category

and that they therefore reside over Western Africa for most of the 10-day period.





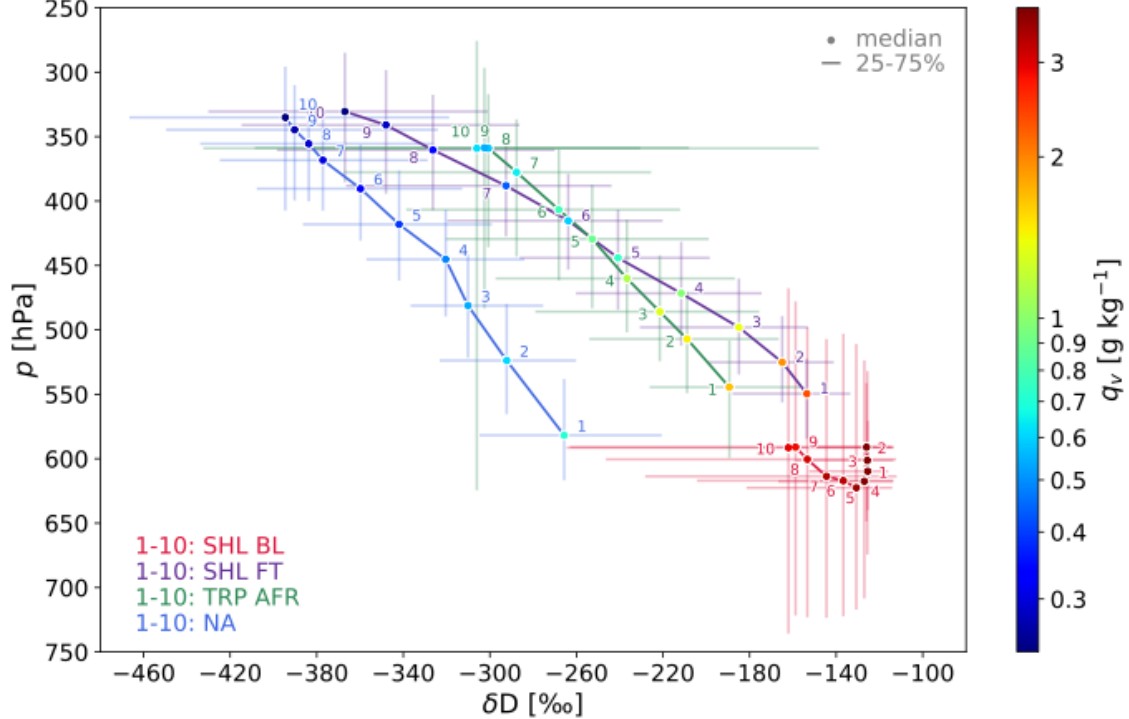

**Figure 9:** $p$–$\delta$D plot summarising the 10-day Lagrangian history of COSMO$_{iso}$ air parcels arriving at 500–700 hPa above Tenerife. The plot shows the median $p$ and $\delta$D over 24 hours by the filled circles and the median $q_v$ of the air parcels by the colours of the filled circles. The numbers indicate the days before the air parcels arrive over Tenerife, where red numbers represent air parcels from the boundary layer of the Saharan heat low (SHL BL), purple numbers air parcels from the free troposphere above the Saharan heat low (SHL FT), green numbers air parcels from tropical Africa (TRP AFR) and blue numbers air parcels from the North Atlantic (NA). The interquartile range of $p$ and $\delta$D is shown by the coloured vertical and horizontal lines.

The TRP AFR transport pathway reflects the typical transport of air parcels that descend from upper-level tropical Africa towards the Canary Islands (Fig. 9). The strong median subsidence ($\Delta p_{10d} = 185$ hPa (10 d)$^{-1}$) over the Sahel and tropical West Africa goes along with a moderate increase in $\delta$D ($\Delta\delta$D$_{10d} = 117$‰ (10 d)$^{-1}$) and $q_v$ ($\Delta q_{v,10d} = 1.1$ g kg$^{-1}$ (10 d)$^{-1}$). In summer, during the West African Monsoon, the Sahel, and therefore air parcels in the TRP AFR transport pathway, are frequently affected by mesoscale convective systems. This implies that moist convective mixing regularly influences air parcels in the TRP AFR transport pathway, in contrast to the SHL BL and SHL FT pathways where dry convective mixing is dominant. The exemplary trajectories presented in Figs. 10e,f show the interplay between subsiding air parcels from the upper-level Sahel and ascending air parcels from the low-level tropical Atlantic and Africa. The low-level air parcels are lifted to higher altitudes thereby experiencing rainout (blue squares in Fig. 10e) and depletion in heavy isotopes. The subsiding air parcels, in turn, get enriched during their subsidence over the Sahel due to mixing with air from lower altitudes. However, the increase in $\delta$D due to this effect is only minor, since the rising lower-level air experiences depletion in heavy isotopes due to condensation in the convective system. The moist convective mixing over the Sahel leads to a homogenisation of the different airstreams, which


subsequently travel over the adjacent tropical Atlantic towards the Canary Islands (Figs. 10e,f). Because of the strong variability in the occurrence of mesoscale convective systems over West Africa, the TRP AFR transport pathway shows high variability (Fig. 9).

Finally, the NA transport pathway presented in Fig. 9 summarises the history of dry air parcels strongly depleted in heavy

isotopes that originate from the upper-level extratropical North Atlantic. This transport pathway leads to the lowest $\delta D$ and $q_v$ values observed in the Canary Islands region. The isotopic composition and humidity only moderately change ($\Delta\delta D_{10d} = 129‰$ $(10\ d)^{-1}$, $\Delta q_{v,10d} = 0.5\ g\ kg^{-1}\ (10\ d)^{-1}$) during the comparatively fast descent ($\Delta p_{10d} = 250\ hPa\ (10\ d)^{-1}$) of the air parcels with low initial $\delta D$ values (median $\delta D = -395‰$) in an environment with regular precipitation events but without deep convective systems. A representative example of the NA transport pathway is given in Figs. 10g,h. The descending air parcels with a very

low $\delta D$ are either directly advected from the upper-level extratropical North Atlantic towards the Canary Islands or, in a few cases, first lifted from the lower-level extratropical North Atlantic to higher altitudes thereby experiencing rainout (blue squares in Fig. 10g) and depletion in heavy isotopes.

Figure 12 summarises the four different transport pathways by showing the spatial distribution of the air parcels' location three days before arrival over Tenerife. Air parcels of the SHL BL transport pathway are almost exclusively located over the Sahara

(Mauretania, Mali, Algeria) before they rapidly move over the adjacent subtropical North Atlantic towards the Canary Islands (Fig. 12a). The direct transport of air parcels from the West African coast towards the Canaries is unique for the SHL BL transport pathway and effectively conserves the isotopic imprint of the dry convective mixing of low-level moisture of diverse origin advected into the SHL. The SHL FT transport pathway, in contrast, has the highest air parcel location frequency over the subtropical North Atlantic south of the Canary Islands and over Mauretania (Fig. 12b). These air parcels typically reside

several days over the subtropical North Atlantic before reaching the Canary Islands from the south. They potentially experience mixing with enriched air over the North Atlantic, which is reflected by the increase in $\delta D$ in water vapour during the last three days before arrival in Fig. 9. Air parcels of the TRP AFR transport pathway are predominantly located over the (sub)tropical North Atlantic three days prior to arrival in the Canary Islands region (Fig. 12c). In addition, they are also found along the West African west coast and over the Sahel. The broad spread in the air parcels' location underlines the large variability of the

TRP AFR regime in terms of pathways of individual trajectories and processes that occur along them. The NA transport pathway also exhibits substantial variability in the air parcels' location over the North Atlantic (Fig. 12d). This variability is presumably closely related to the synoptic variability in the extratropical North Atlantic storm track region and explains the broad $\delta D$ distribution of this category (Fig. 8). The peaks in Fig. 12d imply that air parcels either subside directly from the upper-level extratropical North Atlantic towards the Canary Islands or first travel across the North Atlantic, which often goes

along with precipitation, before descending into the subtropics to the west of the Canary Islands region.





**Figure 10: COSMO$_{iso}$ 10-day backward trajectories showing typical transport pathways of (a,b) air parcels originating from the boundary layer of the Saharan heat low (SHL BL), (c,d) air parcels coming from the free troposphere above the Saharan heat low (SHL FT), (e,f) air parcels originating from tropical Africa (TRP AFR), and (g,h) air parcels coming from the North Atlantic (NA).**
**The trajectories are coloured according to their pressure $p$ (left column) and $\delta$D signal (right column). The black cross depicts the starting position above Tenerife and the black/grey dots the trajectory positions 3/5 days backward in time. Red squares in (a) indicate that a trajectory was in the boundary layer of the Saharan heat low at the respective time and location, purple squares in (c) show that a trajectory was in the free troposphere above the Saharan heat low, and blue squares in (e,g) mark the occurrence of surface precipitation. The time expression in the title indicates the starting time of the backward trajectories.**



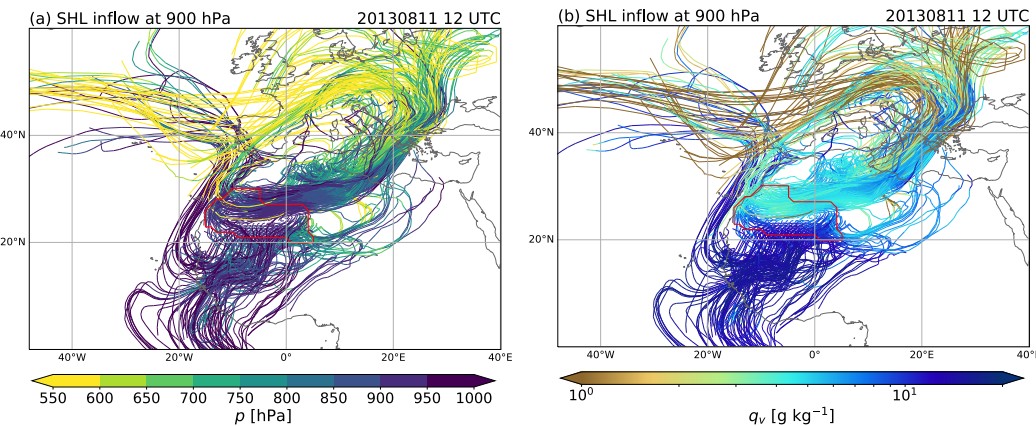

**Figure 11:** Exemplary inflow into the Saharan heat low at 900 hPa. The ERA-Interim trajectories were started every 100 km from the 900 hPa level in the heat low (red contours) at 12 UTC 11 August 2013 and are run 10 days back in time. The starting time corresponds to the time at which air parcels in Figures 10a,b are located in the Saharan heat low (approximately 3 days before arrival). The Lagrangian evolution of the air parcels' pressure $p$ is shown in (a) and of the specific humidity $q_v$ in (b).

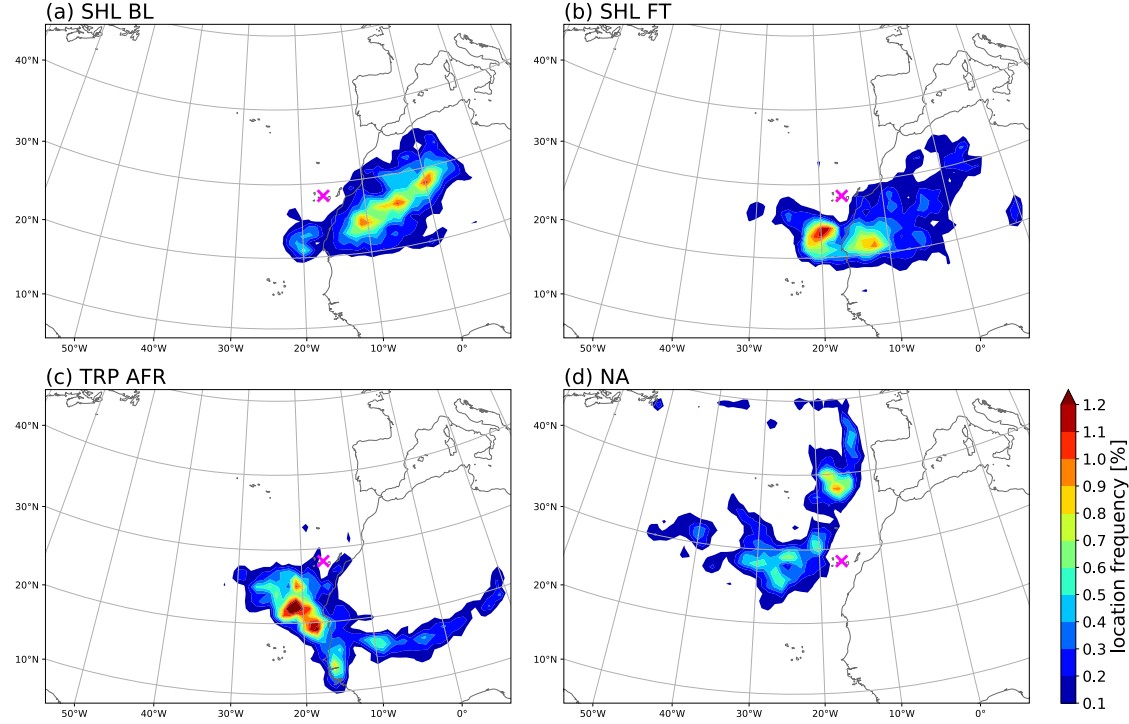

**Figure 12:** Gridded location frequency of COSMO$_{iso}$ backward trajectories three days prior to arrival at 500–700 hPa above Tenerife (magenta cross) classified into groups with different transport pathways. (a) Air parcels originating from the boundary layer of the Saharan heat low (SHL BL, 32% occurrence frequency), (b) air parcels originating from the free troposphere above the Saharan heat low (SHL FT, 26%), (c) air parcels coming from tropical Africa (TRP AFR, 16%) and (d) air parcels originating from the North Atlantic (NA, 26%).



### 4.3 Associated large-scale flow conditions

This section focuses on investigating the large-scale flow conditions associated with the four transport pathways. To this end, we compute ERA-Interim composites of the geopotential height $z$ at 600 hPa and anomalies from the July and August

climatological mean in the period 1979–2018 for each of the four transport pathways. The 6-hourly geopotential height fields in July and August 2013 are assigned to a transport pathway according to the origin of air parcels arriving at 600 hPa above Tenerife in COSMO$_{iso}$ at the corresponding time step. The composites depict the synoptic-scale conditions three days prior to arrival of the respective air parcels. For reasons of consistency, we only consider time steps when COSMO$_{iso}$ and ERA-Interim trajectories both originate either from Africa or from the North Atlantic.

The analysis reveals distinct geopotential height anomaly patterns in the extratropical storm track region for the SHL BL, TRP AFR and NA transport pathways (Fig. 13). The SHL BL transport regime shows a large-scale wave over the North Atlantic with positive $z$ anomalies east of the Canadian east coast and over Europe. The latter has a northeast-southwest tilt and expands to the Canary Islands region (Fig. 13a). In combination with negative anomalies over most of the Sahara, the centre of the North African mid-level anticyclone shifts north-westward to the Moroccan Atlas Mountains. This suggests that extratropical

Rossby wave dynamics influences the export of SHL air to the North Atlantic.

The TRP AFR transport regime also shows a large-scale wave over the North Atlantic with positive $z$ anomalies over the Canadian east coast and over Europe (Fig.13b). In contrast to the SHL BL transport pathway, however, the eastern anomaly is confined to continental Europe. Instead, a negative $z$ anomaly with two centres prevails over the eastern extratropical North Atlantic, one associated with an upper-level trough near 45°N and one southwest of the Canary Islands region, indicative of

an upper-level cutoff. The North African mid-level anticyclone is centred over north-western Algeria, close to its climatological mean position, but with a slightly enhanced amplitude. This dynamical environment enables a direct transport of air parcels from tropical West Africa along the south-western flank of the mid-level anticyclone. The enhanced zonal gradient of $z$ south of the Canary Islands then favours enhanced northward transport towards the Islands. For the SHL FT transport pathway, the geopotential height anomaly pattern (not shown) is similar to that of TRP AFR, albeit with less

pronounced anomalies in the extratropical storm track region and a weaker trough over the central North Atlantic.

By contrast, the NA transport regime is characterised by a pronounced positive $z$ anomaly over the extratropical North Atlantic near the British Isles associated with a ridge to the west and north-west of the Iberian Peninsula as well as a negative anomaly over the Mediterranean Sea (Fig. 13c). These anomalies go along with a clear south-westward shift of the North African mid-level anticyclone over the adjacent subtropical North Atlantic. As a result, African air parcels that are transported westward

with the African Easterly Jet are prevented from reaching the Canary Islands region. The ridge over the extratropical North Atlantic induces subsidence of upper-level air parcels southwards to the Canary Islands. In summary, this analysis demonstrates that extratropical large-scale dynamics influences the mid-tropospheric transport towards the Canary Islands region and thereby the moisture variability.



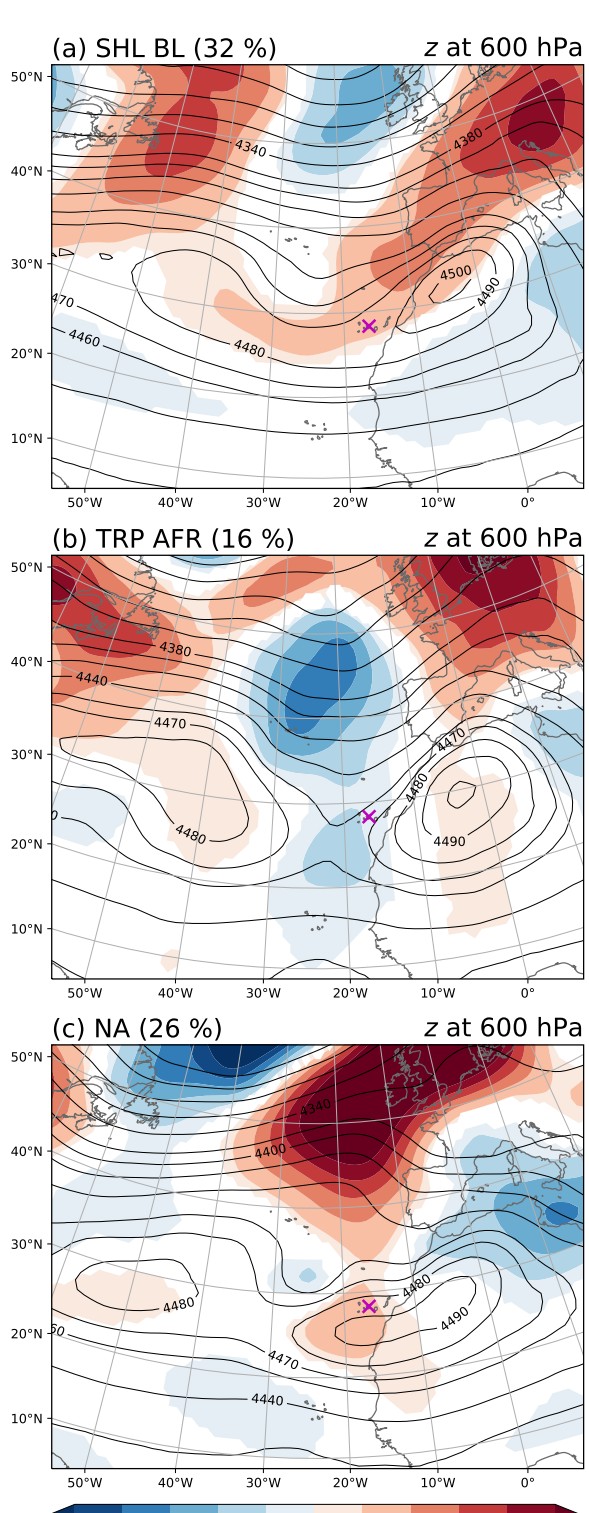

**Figure 13: Anomalies of geopotential height *z* at 600 hPa from the climatological mean of July and August in the period 1979–2018 for three different transport regimes: (a) air parcels originating from the boundary layer of the Saharan heat low (SHL BL), (b) air parcels coming from tropical Africa (TRP AFR), and (c) air parcels originating from the North Atlantic (NA). The classification of the transport pathways relies on the origin of air parcels arriving at 600 hPa above Tenerife. The composites depict the synoptic-scale conditions three days prior to arrival of the respective air parcels. Coloured shadings show significant anomalies (p < 5%) from the climatological mean. Contour lines represent the mean state of the respective transport pathway regime in July and August 2013. The percentages in brackets in the title indicate the occurrence frequency of the transport pathway regime in July and August 2013.**



## 4.4 Climatological perspective

We complete this study by analysing the relevance of the four predominant transport pathways from a climatological perspective. To this end, we compute 10-day ERA-Interim backward trajectories for the period 1979–2018 started every 6 hours above Tenerife. In order to keep the computational costs moderate, the analysis is restricted to the 600 hPa level, which corresponds to the middle of the air layer considered in the previous sections. The trajectories are then grouped into the four different transport pathways (see Sect. 2.4).

The monthly occurrence frequencies of the different transport pathways emphasize the dominant role of the SHL BL and SHL FT categories in July and August (summarised as SHL in Fig. 14). With a median occurrence frequency of about 50% in these two months, SHL clearly exceeds the frequencies of the TRP AFR (~15%) and NA (~33%) transport pathways. The rapid increase of the median occurrence frequency of the SHL transport pathway from 13% in June to 50% in July and the gradual decrease in September and October to 6% are closely related to the SHL onset at the end of June (20 June ± 9 days in the climatological mean; Lavaysse et al., 2009) and its decay in September (17 September ± 7 days). The enhanced transport of mid-tropospheric air from the SHL region in summer develops at the expense of the NA transport pathway, which clearly dominates during the rest of the year. With the onset of the SHL at the end of June, the median occurrence frequency of the NA transport pathway drops from 78% in June to 36% and 30% in July and August, respectively, before reaching again values of around 80% in October. The occurrence frequency of the TRP AFR pathway is less affected by the seasonal evolution of the SHL and varies less in the course of the year (monthly values between 5 and 22%).

From October to May, the NA transport pathway clearly dominates (85% occurrence frequency on average) and alternates with the less frequent TRP AFR transport pathway (13% on average). Note that the TRP AFR transport pathway comprises by definition North African air from outside the SHL (see Sect. 2.4). This implies that in the absence of the SHL, the TRP AFR transport pathway also includes air that originates from the Sahara, unlike in summer, when the SHL is fully established and the TRP AFR transport pathway typically represents air from tropical upper levels south of the SHL (see Sect. 4.2).

The interannual variability of the monthly occurrence frequencies of the SHL, TRP AFR and NA transport pathways is moderate (interquartile ranges in Fig. 14) and amounts to 19%, 9% and 18%, respectively, in July and August (Table 1). In July and August 2013 (studied in detail in this paper), the transport of mid-tropospheric air from the SHL region is more frequent (58%) compared to the 1979–2018 climatology (49%). Specifically, the SHL BL and SHL FT transport pathways both occur about 5% more often (34% and 24%) than in the climatological mean (29% and 20%). The NA transport pathway, by contrast, is underrepresented in July and August 2013 (24%) compared to the climatology (35%). The occurrence of the TRP AFR transport pathway (18%) roughly corresponds to the climatological mean (16%), see also Table 1.

The occurrence frequencies of the four transport pathways in COSMO$_{iso}$ are approximately equal to ERA-Interim in July and August 2013, with differences not exceeding 2% (Table 1). Accordingly, the SHL BL and SHL FT transport pathways appear more often in the time period considered in this study than climatologically, while the NA transport pathway occurs less frequently. However, as the frequencies of the four transport pathways lie within the climatological interquartile ranges, the





results of this study are representative for the contrasting atmospheric conditions that prevail in the Canary Islands region in summer.

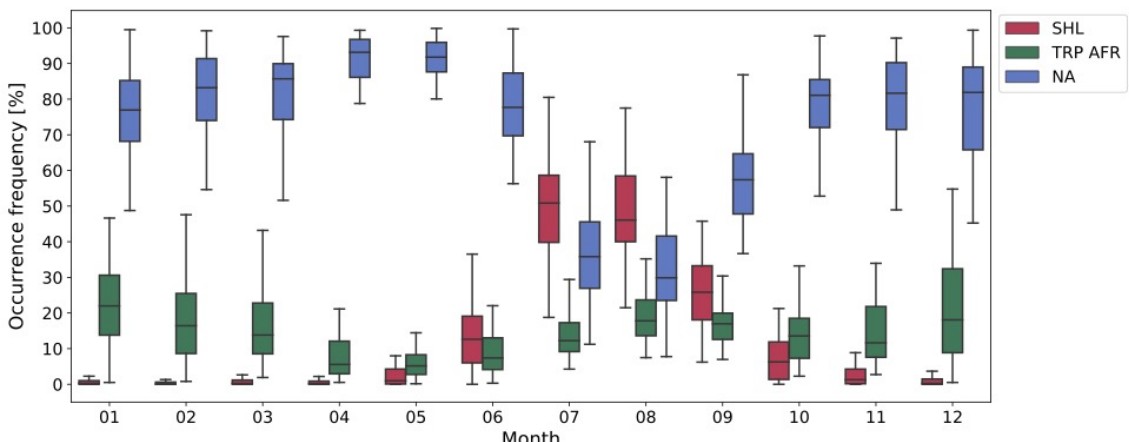

**Figure 14: Monthly climatology of the occurrence frequency of air parcels originating from the convective boundary layer of the Saharan heat low and from the free troposphere above the Saharan heat low (SHL, red boxplots), air parcels coming from tropical Africa (TRP AFR, green boxplots), and air parcels originating from the North Atlantic (NA, blue boxplots). The classification is**
740 **based on backward trajectories started every 6 hours at 600 hPa above Tenerife using ERA-Interim data from 1979–2018. The boxplots show the interquartile range by the extent of the box and the median by the black line in the box. The whiskers correspond to 1.5 times the proportion of the interquartile range past the lower and upper quartiles.**

**Table 1: Occurrence frequency of the four transport pathways defined in this study based on a classification of backward trajectories**
**started every 6 hours from the 600 hPa level above Tenerife. Values correspond to averages and interquartile ranges (in brackets).**

| Dataset and period | SHL (BL + FT) | SHL BL | SHL FT | TRP AFR | NA |
|---|---|---|---|---|---|
| ERA-Interim July–August 1979–2018 | 49% (19%) | 29% (16%) | 20% (8%) | 16% (9%) | 35% (18%) |
| ERA-Interim July–August 2013 | 58% | 34% | 24% | 18% | 24% |
| COSMO$_{iso}$ July–August 2013 | 58% | 32% | 26% | 16% | 26% |

## 5 Summary and conclusions

In this study, we performed a COSMO$_{iso}$ simulation with explicit convection to investigate predominant moisture transport pathways and governing physical processes that affect the free-tropospheric humidity and isotopic variability in the Canary
Islands region in July and August 2013. In addition, we conducted a thorough isotope modelling validation with aircraft and remote sensing observations. The combination of multi-platform isotope observations is an ideal approach to robustly evaluate physical processes in COSMO$_{iso}$ because of the complementary characteristics of the different observational datasets. From the comparison with airborne in situ measurements we learned that COSMO$_{iso}$ reasonably captures the observed variability in



the vertical profiles of specific humidity $q_v$ and $\delta D$ in water vapour, although there is a tendency of overly moist and enriched values in the middle and upper troposphere. These differences can be assigned in the first place to the very heterogeneous meteorological conditions and the large horizontal humidity gradients that occurred during the flights and that are not fully resolved by COSMO$_{iso}$. An overestimation of local mixing processes due to convection by the model may constitute a secondary source of the differences between the model and measurements. Comparing COSMO$_{iso}$ to ground-based remote sensing observations from the NDACC/FTIR site in Tenerife further demonstrated that our simulation can reproduce the observed day-to-day variability of mid-tropospheric $q_v$ and $\delta D$, where daily values vary from 0.45 g kg$^{-1}$ to 4.3 g kg$^{-1}$ for $q_v$ and from $-318$‰ to $-128$‰ for $\delta D$. The statistical validation against satellite-based remote sensing from the IASI sensor observations confirmed the good agreement between the modelled and observed $q_v$ and $\delta D$ values in the middle troposphere, apart from a slightly negative $\delta D$ bias of COSMO$_{iso}$ compared to the satellite data. Due to the complex characteristics of the remote sensing observations and the simulated averaging kernels that are applied to the model data, however, it is challenging to identify the reasons behind these biases. Overall, the comparison of COSMO$_{iso}$ with multi-platform isotope observations highlighted that COSMO$_{iso}$ is able to reproduce the observed variations in $q_v$ and $\delta D$ in the middle troposphere and can thus be used to study the isotopic variability that occurs on time scales of hours to days over the eastern subtropical North Atlantic. The good performance of COSMO$_{iso}$ in representing the isotopic variability allowed us to thoroughly investigate the atmospheric processes behind the observed short-term isotope variations in the subtropical free troposphere in the layer between 500 and 700 hPa. Based on an analysis of isotope signals along COSMO$_{iso}$ 10-day backward trajectories, we showed that this variability can be linked to four different transport pathways, each associated with a distinct isotope signature:

(i)     Most humid and enriched (median $\delta D = -122$‰) conditions in the Canary Islands region appear during the transport of North African air from the convective boundary layer of the Saharan heat low (SHL; referred to as SHL BL, 32% occurrence frequency in July and August 2013). Two contrasting airstreams typically add to this transport pathway: subsiding air parcels from the upper-level extratropical North Atlantic and low-level air parcels with a high $\delta D$ from eastern North Africa. Both airstreams converge in the SHL, where dry convective mixing effectively homogenises the contrasting isotopic compositions, and subsequently travel as a well-mixed air mass towards the Canary Islands. The exceptionally homogeneous and enriched isotopic composition of this air mass is a unique characteristic of the SHL BL transport pathway and a clear imprint of the dry convective mixing in the SHL.

(ii)     The transport of North African air that originates from the free troposphere above the SHL (SHL FT, 26% occurrence frequency) also leads to high $\delta D$ values (median $\delta D = -148$‰) in the Canary Islands region. However, the isotopic composition of these air parcels is a bit less enriched in heavy isotopes and more variable compared to the SHL BL air. This transport pathway typically represents initially very dry air parcels with a very low $\delta D$ that experience a strong increase in $q_v$ and $\delta D$ over the SHL region during their subsidence from the upper troposphere. This enrichment in heavy isotopes in the free troposphere over the SHL presumably results from mixing in of high-$\delta D$ air injected



from the boundary layer of the SHL and notably represents the largest change in humidity and isotopic composition among the four transport pathways.

(iii)    With a median $\delta D$ signal of –175‰, North African air originating from outside the SHL region is more depleted in heavy isotopes than air of the SHL BL and SHL FT transport pathways. This category (TRP AFR, 16% occurrence frequency) reflects the transport of air parcels that subside from upper-level tropical Africa towards the Canary Islands. During their transport over Sahelian and tropical West Africa, where mesoscale convective systems often occur in summer, these air parcels experience only a moderate moistening and enrichment by moist convective mixing since frequent cloud formation and rainout lead to a depletion in heavy isotopes in water vapour. Due to the considerable variability in the occurrence and characteristics of convection, the TRP AFR category shows high variability in terms of pathways of individual trajectories.

(iv)    The driest and most depleted (median $\delta D$ = –255‰) conditions in the Canary Islands region appear during transport of air from the North Atlantic (NA, 26% occurrence frequency). This transport pathway is characterised by a strong subsidence of initially very dry air parcels with a very low $\delta D$ from the upper-level extratropical North Atlantic, which goes along with a moderate enrichment in heavy isotopes and slight moistening due to mixing with lower-level air parcels. There is, however, substantial variability in terms of pathways of individual trajectories, which is reflected in the large variability of the isotopic composition, and which is presumably related to the synoptic-scale flow variability along the extratropical North Atlantic storm track region.

We further demonstrated that each of the four different transport pathways is associated with specific large-scale flow anomalies. Specifically, distinct geopotential height anomaly patterns appear in the extratropical storm track region for each transport pathway. The SHL BL transport pathway is characterised by a positive geopotential height anomaly over Europe and the eastern North Atlantic. This anomaly coincides with a strong shift of the North African mid-level anticyclone to the north-west over the Atlas Mountains, which promotes the subsidence of upper-level extratropical air into the convective boundary layer of the SHL. By contrast, the TRP AFR transport pathway is associated with a distinct negative geopotential height anomaly and an upper-level trough over the eastern extratropical and subtropical North Atlantic. This favours the transport of air parcels from tropical West Africa along the south-western flank of the North African mid-level anticyclone towards the Canary Islands region. The SHL FT transport pathway shows similar geopotential height anomalies as TRP AFR. Finally, in the NA transport regime, a pronounced positive geopotential height anomaly and a strong upper-level ridge prevail over the extratropical North Atlantic, which goes along with a strong westward shift of the North African mid-level anticyclone over the adjacent subtropical North Atlantic. This dynamical environment, on the one hand, induces subsidence of extratropical upper-level air parcels towards the Canary Islands, and on the other hand, prevents North African air parcels from travelling towards the Canary Islands region. Overall, the results emphasise that the extratropical large-scale circulation notably influences the air parcel transport pathways over the subtropical eastern North Atlantic.

A climatological analysis of the transport pathways in the period of 1979–2018 highlights the importance of the SHL transport pathways in summer, and their complete absence in the extended cold season. With the climatological onset of the SHL in





June, the SHL transport pathways become increasingly important and together dominate in July and August (29% and 20% occurrence frequency of SHL BL and SHL FT, respectively) over the TRP AFR (16%) and NA (35%) transport pathways before becoming less relevant in September when the SHL decays. The NA transport pathway dominates from October to May in the absence of the SHL. In July and August 2013 (studied in detail in this paper), the occurrence frequencies of the four transport pathways lie within the climatological interquartile ranges. We may speculate that the interannual variability in these

occurrence frequencies in summer is related to the interannual variability in the intensity of the SHL and the West African monsoon. Given the distinct differences in the isotopic composition of the four transport pathways, water vapour isotopes might thus be regarded as an integral measure of West African dynamics.

In this study, we focused on the vertical layer between 500 and 700 hPa, since the remote sensing observations, against which we evaluated our COSMO$_{iso}$ simulation, have the highest sensitivity at these altitudes. In addition, the humidity and isotopic

composition in this layer are predominantly influenced by the large-scale circulation. However, local mixing processes between the marine boundary layer and the free troposphere are still potentially important for the subtropical free-tropospheric moisture budget, in particular in the vicinity of islands where thermally driven upslope flows weaken the strong inversion at the boundary layer top (e.g., Bailey et al., 2013; González et al., 2016), and represent an interesting aspect for future studies. Another potential limitation of this study represents the sensitivity of the SHL BL and SHL FT transport pathway definitions

to the boundary layer height, which is difficult to diagnose and thus subject to uncertainties (e.g., Marsham et al., 2013a; Engelstaedter et al., 2015; Garcia-Carreras et al., 2015). Yet, since the occurrence frequencies of the SHL BL and SHL FT transport pathways in July and August 2013 only differ by 2% between COSMO$_{iso}$ and ERA-Interim, we are confident that the uncertainty in the boundary layer height is of minor importance. Furthermore, also with explicitly resolved convection, simulations of the West African Monsoon still have substantial biases (Marsham et al., 2013b; Pante and Knippertz, 2019), for

instance due to challenges associated with the correct representation of mesoscale convective systems. Hence, uncertainties in the representation of these systems might be reflected in the transport of North African air and consequently in the simulated isotopic composition in the free troposphere above the Canary Islands region. Finally, we acknowledge the limited ability of the adopted trajectory approach to represent mixing processes. Since the temporal and spatial resolution of the backward trajectories is beyond the spatiotemporal scales of convective and turbulent mixing processes, mixing can be only indirectly

deduced from the trajectories. A future study on the relative importance of different moisture sources for the free-tropospheric humidity and isotopic composition over the (sub)tropical North Atlantic and West Africa is planned, using passive tracers following water that evaporates from specific source regions throughout the simulation (the so-called tagging technique; Koster et al., 1986; Winschall et al., 2014).

In summary, this paper presents a comprehensive isotope modelling validation on hourly to weekly time scales with aircraft

and remote sensing observations, provides new insights into the process history of the prevailing moisture transport pathways in the free troposphere over the eastern subtropical North Atlantic, and constitutes a sound framework for the interpretation of observed short-term variations in humidity and water vapour isotopes in this region.





**Appendix A: Overview of mid-tropospheric moisture and wind conditions during the MUSICA aircraft campaign in July and August 2013**

In the paper, we show that differences in specific humidity $q_v$ and $\delta D$ in water vapour between COSMO$_{iso}$ and airborne in situ measurements can be partly explained by synoptic-scale uncertainties in COSMO$_{iso}$, which are associated with the very heterogeneous meteorological conditions that occurred during the summer 2013 MUSICA aircraft campaign. Figures A1–A7 below indicate the heterogeneous moisture conditions and large $q_v$ and $\delta D$ gradients in the middle troposphere near the Canary Islands in the eastern subtropical North Atlantic. The large-scale moisture distribution and atmospheric flow situation in

COSMO$_{iso}$, however, agrees well with ERA-Interim reanalysis data (Figs. A1–A7).



**Figure A1: ERA-Interim and COSMO$_{iso}$ moisture fields (coloured shading in a, c, e) and horizontal wind fields (coloured shading in b, d) at 500 hPa at 12 UTC 21 July 2013. Panel (a) shows the specific humidity $q_v$ in ERA-Interim, (b) the horizontal wind in ERA-Interim, (c) the specific humidity $q_v$ in COSMO$_{iso}$, (d) the horizontal wind in COSMO$_{iso}$, and (e) $\delta$D in water vapour in COSMO$_{iso}$. The black contours in the left panels depict the geopotential height at 500 hPa and the green arrows in the right panels the horizontal wind direction.**





**Figure A2: Same as Fig. A1 but for 12 UTC 22 July 2013.**



**Figure A3: Same as Fig. A1 but for 12 UTC 24 July 2013.**





**Figure A4: Same as Fig. A1 but for 12 UTC 25 July 2013.**





**Figure A5: Same as Fig. A1 but for 12 UTC 30 July 2013.**



**Figure A6: Same as Fig. A1 but for 12 UTC 31 July 2013.**





Figure A7: Same as Fig. A1 but for 12 UTC 1 August 2013.





## Appendix B: Comparison of a COSMO$_{iso}$ simulation with parametrised convection to multi-platform isotope observations

In the paper, we compare a two-month COSMO$_{iso}$ simulation with explicit convection of July and August 2013 to multi-platform water vapour isotope observations in order to quantitatively evaluate the performance of COSMO$_{iso}$ in modelling the

free-tropospheric variability of humidity and isotopic composition on time scales of hours to days in contrasting atmospheric conditions over the eastern subtropical North Atlantic. We also performed a simulation with the same horizontal resolution (0.125°) and with parameterised convection, which however led to larger model biases in comparison with airborne, ground- and space-based observations (Fig. B1).



**Figure B1: Comparison of COSMO$_{iso}$ $q_v$ and $\delta$D in water vapour from a convection-parameterised simulation with multi-platform observations over the subtropical North Atlantic near Tenerife. (a-c) comparison with airborne in situ ISOWAT measurements performed between sea level and 7 km altitude during 7 flight days in July and August 2013, (d-f) comparison with satellite-based IASI remote sensing observations retrieved for 4.2 km in a 10° x 10° box centred around Tenerife during 32 days in July and August 2013, (g-i) comparison with ground-based FTIR remote sensing observations retrieved for 4.9 km during 25 days in July and August 2013. Solid/dashed contours in panels (d-f) show the 50/90% frequency isolines of normalised two-dimensional histograms. Empty circles in panels (g-i) indicate a less reliable air parcel transport (see Sect. 2.4). The black box in panel (b) represents the $\delta$D space of panels (e,h).**



*Author contributions.* MS provided the stable water isotope data measured during the MUSICA ISOWAT campaign and the ground-based remote sensing data (MUSICA NDACC/FTIR). MS, CD and BE provided the satellite-based remote sensing observations (MUSICA IASI) and performed the post-processing of the COSMO$_{iso}$ data used for the comparison with MUSICA IASI. MW provided the ECHAM5-wiso boundary data that was used for the COSMO$_{iso}$ simulations. FD performed the COSMO$_{iso}$ simulations, the model evaluation against observations and the trajectory-based analysis. FD wrote the paper, with regular input from HW, FA and SP. All co-authors contributed to the interpretation of the results and commented on the manuscript.

*Competing interests.* The authors declare that they have no conflict of interest.

*Acknowledgements.* We acknowledge funding from the German-Swiss project "MOisture Transport pathways and Isotopologues in water Vapour (MOTIV)" supported by the Swiss National Science Foundation Grant No. 164721 and the Deutsche Forschungsgemeinschaft under the project ID 290612604. The COSMO$_{iso}$ simulations were performed at the Swiss National Supercomputing Centre (CSCS) with the small production projects sm08 and sm32. Further post-processing of the COSMO$_{iso}$ simulations and the MUSICA IASI retrievals were performed at the supercomputer ForHLR funded by the Ministry of Science, Research and the Arts Baden-Wuerttemberg and by the German Federal Ministry of Education and Research. This work strongly benefits from the project MUSICA (funded by the European Research Council under the European Community's Seventh Framework Programme (FP7/2007-2013)/ERC Grant Agreement number 256961). The authors acknowledge MeteoSwiss and ECMWF for the access to the ERA-Interim reanalyses.

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
