# Peer review of "Disentangling different moisture transport pathways over the eastern subtropical North Atlantic using multi-platform isotope observations and high-resolution numerical modelling"

_Atmospheric Chemistry and Physics, 2021_

## Referee Comment (RC2)

**Review of the article by Dahinden et al**

July 11, 2021

This study presents an evaluation of an isotope-enabled regional model simulation over Canary Islands compared to airborne, ground-based remote-sensing and satellite observations. Using this simulation, it investigates the variability in humidity and isotopic composition using back-trajectories. The main result is that humidity and isotopic composition mainly depends on the origin of air masses, which is associated with synoptic weather patterns.

The article is well-organized, well-written, well-illustrated. The rationales are sound. The methods are extensively described.

This study will be interesting for the people in the water isotope community. It provides a methodological framework for investigating observed synoptic variability in isotopic composition. It adds to the studies showing the importance of synoptic-scale weather patterns in controlling the humidity and isotopic composition in subtropical regions.

Apart from more discussion of the results with respect to previous studies, I recommend mostly minor revisions.

**1 General comment: discuss more the results with respect to previous studies**

More discussion of the results would be valuable. A discussion section could be added before the conclusion. This discussion section could also help make the conclusion section more concise by moving some paragraphs from the conclusion to the discussion.

- The study by [Lacour et al., 2017] was on a very similar topic. It is cited in the introduction but never after. It diserves to be more discussed. To what extent are your results consistent with their study? What is the added value of this study compared to their study? What new do we learn?

- Section 4.3 on large-scale flow has no citation. I would be surprised that none has ever investigated the origin of air masses and synoptic weather patterns in this region. How do your results compare with previous studies? What is the added value of this study relative to the state of the art?

- The introduction advertises about the added value of water isotopic observations. What do we learn in this study that we couldn't learn without isotopic observations? Couldn't the back-trajectory analysis and the $q_v$ along trajectories alone be sufficient? The isotopic simulation is useful to interpret the isotopic observations, but is there any usefulness of isotopic observations or simulations for people beyond the isotopic community? A few examples:

  - l 500: "added value of water vapor isotopes": what would fig 8a look like with $q_v$? Wouldn't we get distinct signature as well? Fig 8b suggests that $q_v$ and $\delta D$ are strongly correlated...
  - fig 9: would it be insightful to show these trajectories in a $q_v$- $\delta D$ plot?
  - l827: Can't $q_v$ be similarly regarded as an integral measure of West AFrican Dynamics?

  ⇒I suggest to add some discussion about to what extent water isotopic observations provide an added value compared to $q_v$ , or not.

**2 Minor comments**

- l 116: Blossey et al 2010 did not use a regional circulation model and did not study synoptic-scale variability. It was a cloud resolving model (higher resoution) in a very idealized setting that allows the simulation of the tropical circulation in a stationary state. I wouldn't list it with the other studies here.

- l 228: can you give more details about your cloud filtering? e.g. what threshold, what altitude for the cloud fraction? Is the fraction of selected scenes in the model similar to this fration in reality? This could go in an appendix.

- l 249: same here: can you give more details about your cloud filtering? Why is it different from that for FTIR? What thresholds do you use? Is the fraction of selected scenes in the model similar to this fration in reality? This could go in the same appendix as above.

- l 252-253: what does this mean? Variability of what? Of the averaging kernels? Or of $\delta D$? Do you mean retrieval simulator combined with COSMOiso, or is it an intrisic property of te simulator?

- l 323: I didn't follow why there are only seven trajectories. I thought there were seven days with many trajectories for each day?

- l 343-344: it must be clarified here that most of the apparent variability is associated with the vertical gradient in humidity and $\delta D$. "short-term variability" is confusing here because it evokes temporal variability, whereas here the vertical variability is probably dominant.

- l 345-347: "$\Delta ln(q_v)$ " -> "$\Delta q_v$", since the values are in g/kg. The values for $\Delta ln(q_v)$ would have no unit, or in %.

- l 351: does this sampling includes vertical variations? If so, clarify that the shading probably mainly reflects the vertical variations?

- l 364: I don't understand this rationale. Why couldn't be the over-estimated $q_v$ and $\delta D$ due to the dry and depleted tongue around 15W that does not reach far enough towards the equator (around 30N in COSMO and 25N in ERAI)?

- l 411: "despite small" or "in spite of small"

- l 421: Couldn't there be an enriched bias for $\delta D$ observed by FTIR, and to a lesser extent by IASI? The comparison with in-situ data suggests that the model has an enriched bias, not a depleted bias. Previous studies cross-comparing different datasets have shown that the FTIR observations are often the most enriched ([Risi et al., 2012, Lacour et al., 2015]).

- Are the COSMOiso simulations and observations used in the comparison available in a repository?

**References**

[Lacour et al., 2015] Lacour, J.-L., Clarisse, L., Worden, J., Schneider, M., Barthlott, S., Hase, F., Risi, C., Clerbaux, C., Hurtmans, D., and Coheur, P.-F. (2015). Cross-validation of iasi/metop derived tropospheric ?d with tes and ground-based ftir observations. *Atmospheric Measurement Techniques*, 8(3):1447–1466.

[Lacour et al., 2017] Lacour, J.-L., Flamant, C., Risi, C., Clerbaux, C., and Coheur, P.-F. (2017). Importance of the saharan heat low in controlling the north atlantic free tropospheric humidity budget deduced from iasi $\delta$d observations. *Atmospheric Chemistry and Physics*, 17:9645–9663.

[Risi et al., 2012] Risi, C., Noone, D., Worden, J., Frankenberg, C., Stiller, G., Kiefer, M., Funke, B., Walker, K., Bernath, P., Schneider, M., Wunch, D., Sherlock, V., Deutscher, N., Griffith, D., Wernberg, P., Bony, S., Jeonghoon Lee, D. B., Uemura, R., and Sturm, C. (2012). Process-evaluation of tropical and subtropical tropospheric humidity simulated by general circulation models using water vapor isotopic observations. Part 1: model-data intercomparison. *J. Geophy. Res.*, 117:D05303.

---

## Author Response (AR1)

**Answers to the reviewers' comments**

We thank the two reviewers for their insightful comments. Below, we address each comment point by point. The reviewers' comments are repeated in blue, our responses are given in black and the changes to the manuscript in black italic.

**Reviewer 1**

**General Comments**

The paper investigates different moisture pathways to the Canary Islands using a multi-platform approach that is based on water isotopes. The use of an isotope-enabled numerical model, which is carefully validated against observations, is quite an innovative technique and allows great insights into the origin of the air parcels that reach the Canary Islands. The paper is overall well written, and the results are significant. I think it should be published following minor reviews.

We thank the reviewer for this positive general feedback.

**Minor Comments**

1) Are the different pathways associated with different weather patterns (e.g., cold front vs. trade wind shower)? If so, maybe this should be stated clearly and discussed a bit.

Thanks for your interest in the surface weather associated with the different pathways. Although this aspect is not central for our study, which focuses on the variability of humidity in the mid-troposphere, we analysed the statistical distributions of 2-m temperature, $T_{2m}$, for the four pathways (see Fig. R1 in this reply document). Generally, due to the location of Tenerife on the south-eastern side of the Azores' high-pressure system, surface weather conditions are rather stationary in summer. Variations of $T_{2m}$ are small and the distributions for the four pathways reveal median differences of about 1 K between the pathways from the North Atlantic (NA) and from the Saharan heat low boundary layer (SHL BL). Other weather parameters also show very little variation during summer (e.g., no precipitation occurred in July and August 2013). Fronts are very rare in this region, in particular in summer (Fig. 5 in Schemm et al., 2015). Surface weather variations would be larger during the other seasons, but an analysis of these other seasons is clearly outside the scope of this study.

[Figure]

*Fig R1: Boxplots of hourly COSMO$_{iso}$ 2-m temperatures ($T_{2m}$) sampled in a 5° x 5° box centred around Tenerife (see orange box in Fig. 1 in the manuscript), separated for the four different transport regimes studied in the paper: air parcels originating from the upper-level extratropical North Atlantic (NA, 26% occurrence frequency), from tropical Africa (TRP AFR, 16%), from the upper levels above the Saharan heat low (SHL FT, 26%) and from the Saharan heat low (SHL BL, 32%). The classification of the transport pathways relies on the origin of air parcels arriving at 600 hPa above Tenerife (as explained in the manuscript). The boxplots show the interquartile range by the extent of the box and the median by the black line in the box. The whiskers correspond to 1.5 times the proportion of the interquartile range past the lower and upper quartiles.*

2) Could you give some more reasons as to why certain pathways in the period you examined occur more/less frequently than the climatology? For instance, can you link this to large-scale modes like the NAO? You hint at this around line 825, but it would be nice to see a more in-depth discussion.

Thank you for this comment, it is a good suggestion to discuss the interannual variability of the different transport pathways in more depth. In the revised version, we address possible reasons for the anomalously frequent transport of mid-tropospheric air from the SHL region (58%) in July and August 2013 compared to the 1979–2018 climatology (49%) and the less frequent occurrence of the NA transport regime (24%) compared to the climatology (35%) in Sect. 4.4:

*p. 30, l. 772–781: "We analysed the relationship between teleconnection indices[1] and the occurrence frequency of the transport pathways for July and August in the period 1979–2018. The North Atlantic Oscillation (NAO) index is weakly correlated with the occurrence frequency of the SHL and NA transport pathways (Pearson correlation coefficients r = 0.13 and –0.14, respectively). The Multi-variate ENSO Index (MEI) shows a moderate correlation with the occurrence frequency of the SHL and NA regimes (r = 0.34 and –0.37, respectively). The frequency of the TRP AFR transport does not correlate with NAO nor with MEI. These correlations indicate that the anomalously frequent transport of air from the SHL region in July and August 2013 is most likely linked to the anomalies in the NAO (2.52 in July and 2.16 in August) and MEI (–0.5 in both months). This is in qualitative agreement with Rodríguez et al. (2015), who found a negative correlation between the MEI and Saharan dust concentrations at the Izaña observatory on Tenerife, which in turn is a measure for the transport of air masses from the SHL (González et al., 2016)."*

**Technical Comments**

- Line 24: "and thus allows" → "thus allowing"

Changed according to the reviewer's suggestion.

- Line 39: "large scale-flow" → "large-scale flow"

Thank you, done.

- Line 60: Just to make the paper a bit more self-contained, please define "Intertropical Discontinuity"

There is now a short note in the manuscript:

*p. 2, l. 62–64: "The Intertropical Discontinuity describes a sharp air mass boundary at about 20°N that is characterised by large contrasts in humidity, temperature, and vertical stability (Fink et al., 2017)."*

- Line 421: "preliminary" → "mainly"

Changed according to the reviewer's suggestion.

- Line 434: "alter" → "change"

Done.

- Fig 7: Is there anything that could be learnt from deuterium excess?

With respect to the different transport regimes, the deuterium excess does not provide additional information as can be seen in Fig. R2. However, the deuterium excess shows several interesting short-term variability patterns in the boundary layer and around 500 hPa. In particular, surprisingly low deuterium excess values ($d$ < 10‰) are simulated by COSMO$_{iso}$ in the mid troposphere. However, an attribution of these signals to individual processes is not the focus of this paper, mainly because we do
* * *
[1] Data downloaded from the NOAA Physical Sciences Laboratory: https://psl.noaa.gov/data/climateindices/list/

not have remote sensing observations to validate them. We therefore do not show or discuss any $\delta^{18}O$ or deuterium excess signals in the paper.

[Figure]

*Fig. R2: COSMO$_{iso}$ (a) $\delta D$ and (b) deuterium excess $d$ in water vapour 300–900 hPa above Tenerife, as well as (c) the Lagrangian origin of the air parcels. The black horizontal lines confine the horizontal air layer between 500–700 hPa, which is the focus of this study. Black dots in (a,b) indicate a less reliable air parcel transport (see Sect. 2.4). Black dots in (c) represent transport pathways with surface precipitation equal or larger than 1 mm h$^{-1}$ at least once over continental Africa for the African air parcels and over the North Atlantic for the North Atlantic air parcels, respectively.*

- Line 777: "travel" → "travels"

Thank you, done.

- Line 814: "This dynamical environment, on the one hand" → "On the one hand, this dynamical environment"

Changed according to the reviewer's suggestion.

**Reviewer 2**

This study presents an evaluation of an isotope-enabled regional model simulation over Canary Islands compared to airborne, ground-based remote-sensing and satellite observations. Using this simulation, it investigates the variability in humidity and isotopic composition using back-trajectories. The main result is that humidity and isotopic composition mainly depends on the origin of air masses, which is associated with synoptic weather patterns.

The article is well-organized, well-written, well-illustrated. The rationales are sound. The methods are extensively described.

This study will be interesting for the people in the water isotope community. It provides a methodological framework for investigating observed synoptic variability in isotopic composition. It adds to the studies showing the importance of synoptic-scale weather patterns in controlling the humidity and isotopic composition in subtropical regions.

We thank the reviewer for this overall positive feedback.

**General Comments**

More discussion of the results would be valuable. A discussion section could be added before the conclusion. This discussion section could also help make the conclusion section more concise by moving some paragraphs from the conclusion to the discussion.

Thank you for this comment. We agree with the reviewer and expanded the discussion of our work in the results sections 3 and 4. Furthermore, we put the conclusions in a more concise form.

1) The study by Lacour et al. (2017) was on a very similar topic. It is cited in the introduction but never after. It deserves to be more discussed. To what extent are your results consistent with their study? What is the added value of this study compared to their study? What new do we learn?

Thanks for pointing out this deficiency in the original manuscript. It is true that the studies by Lacour et al. (2017) and González et al. (2016) are on a similar topic. Both studies investigated moisture transport pathways to the subtropical North Atlantic free troposphere based on isotope observations (IASI and ground-based in situ, respectively) and backward trajectories using reanalysis data. They found that moist, isotopically enriched air is associated with transport from North Africa, whereas dry, depleted air originates from the extratropical North Atlantic. Lacour et al. (2017) mainly focused on the seasonal and interannual variability of the isotope composition observed above the Canary Islands and showed that it is closely linked to the activity of the SHL. Our study, which combines airborne, ground- and space-based observations of $\delta D$ with high-resolution simulations, confirms these results. Additionally, we highlight the large variability on the synoptic time scale and investigate the reasons for this variability with a detailed analysis of isotope signals along COSMO$_{iso}$ backward trajectories from the Canary Islands. This enables us to directly link observed $\delta D$ signals to the origin of moisture and to disentangle involved physical processes. In particular, we demonstrate that North African air masses affected by dry convective mixing in the SHL region and air masses influenced by moist convection in the Sahel region further south are associated with a distinct isotope signature. We further show that the different moisture transport pathways defined in this study are related to specific large-scale flow conditions and extratropical Rossby wave dynamics. Overall, the combination of the isotope-enabled model COSMO$_{iso}$ with the Lagrangian diagnostics and the multi-platform water vapour isotope observations provides a solid framework to analyse and explain the observed atmospheric isotope signals beyond simple Rayleigh distillation and mixing models. In turn, the high-resolution isotope observations allow a robust evaluation of physical processes in the model, which are difficult to constrain by measurements of specific humidity alone.

We now discuss these aspects in sections 4.1 and 4.4:

*p. 20, l. 524–534: "This analysis confirms the current state of knowledge about the contrasting moisture transport conditions over the eastern subtropical North Atlantic, resulting from an alternation of humid, isotopically enriched air primarily coming from Africa and of dry, depleted air mainly originating from the upper-level extratropical North Atlantic (González et al., 2016; Lacour et al., 2017). In addition, our work shows that North African air masses affected by the SHL (pathways SHL BL and SHL FT) and air masses originating from the Sahel region further south (pathway TRP AFR) are associated with a distinct isotope signature. The combination of high-resolution numerical isotope modelling with multi-platform isotope observations, which represents an expansion of the previous observation-oriented studies by González et al. (2016) and Lacour et al. (2017), offered the possibility to directly link the observed synoptic time scale variability of specific humidity and isotope composition to the origin of moisture. In particular, it allows for studying the isotopic composition along backward trajectories from the Canary Islands region and thereby disentangling the governing physical processes that affect the subtropical free-tropospheric moisture budget (see Sect. 4.2)."*

*p. 30, l. 754–755: "This is in agreement with Lacour et al. (2017), who showed that the seasonality in the transport of air from the SHL region is linked to the SHL activity."*

2) Section 4.3 on large-scale flow has no citation. I would be surprised that none has ever investigated the origin of air masses and synoptic weather patterns in this region. How do your results compare with previous studies? What is the added value of this study relative to the state of the art?

We agree with the reviewer and added more discussion of our results about the large-scale flow situation in the context of previous studies in Sect. 4.3. Specifically, we have made the following changes to the manuscript:

a) Overview of previous studies on the interactions between extratropical Rossby wave dynamics and synoptic circulations in north-western Africa:

*p. 27, l. 667–674: "Previous work has shown that even during the warm season the investigation region is frequently affected by positively tilted upper-level intrusions of high potential vorticity (PV) from higher latitudes, usually associated with Rossby wave breaking over the North Atlantic (Fröhlich and Knippertz, 2008; Papin et al., 2020). Ahead of these troughs, moist mid-level air can be transported northwards around the western flank of the anticyclone overlaying the low-level SHL and lead to precipitation in north-western Africa (Knippertz et al., 2003; Knippertz, 2003). Upper-level troughs and ridges together with African easterly waves determine synoptic-scale fluctuations of the SHL and its associated mid- to upper-level anticyclone (Lavaysse et al., 2010) as well as of dust export from Africa to the Atlantic (Cuevas et al., 2017)."*

b) Discussion of our results about large-scale flow conditions and subtropical mid-tropospheric air transport in the context of previous studies:

*p. 27, l. 684–688: "In combination with negative anomalies over most of the Sahara, the centre of the North African mid-level anticyclone shifts north-westward to the Moroccan Atlas Mountains such that Saharan air can reach the Canary Islands flowing along its southern flank. Cuevas et al. (2017) referred to this as a high NAFDI (North African Dipole Intensity) index situation. This confirms previous work showing that extratropical Rossby wave dynamics can influences the export of SHL air to the North Atlantic."*

*p. 27, l. 694–698: "This dynamical environment enables a direct transport of air parcels from tropical West Africa along the south-western flank of the mid-level anticyclone similar to the situation described in Knippertz et al. (2003) and Knippertz (2003). The enhanced zonal gradient of z south of the Canary Islands then favours intensified northward transport towards the islands. Cuevas et al. (2017) referred to this as a low NAFDI index situation."*

*p. 28, l. 708–711: "In summary, this analysis is in agreement with previous work that extratropical Rossby wave dynamics strongly influences the mid-tropospheric transport in the study region and thereby the moisture and isotope variability. Specifically for the Canary Islands, the position and zonal extent of the anticyclone above the SHL determines the influx of air from tropical Africa, the Sahara or the North Atlantic."*

3) The introduction advertises about the added value of water isotopic observations. What do we learn in this study that we couldn't learn without isotopic observations? Couldn't the back-trajectory analysis and the qv along trajectories alone be sufficient? The isotopic simulation is useful to interpret the isotopic observations, but is there any usefulness of isotopic observations or simulations for people beyond the isotopic community? A few examples:

Thank you for this remark, we agree that many signals in $\delta$D can also be seen in the specific humidity $q_v$ but we think that our study clearly shows the additional value of the stable water isotopes. In particular, we demonstrate that isotope signals allow for discriminating between the three different African transport regimes, which would not be possible by means of humidity measurements alone (see explanation below and Fig. 8b in the manuscript).

– l 500: "added value of water vapor isotopes": What would Fig. 8a look like with $q_v$? Wouldn't we get distinct signature as well? Fig. 8b suggests that $q_v$ and $\delta$D are strongly correlated...

The statistical analysis of the mid-tropospheric $\delta$D signal highlights that each of the four transport pathways has a distinct isotopic signature (Fig. R3 left panel). This is not the case for specific humidity $q_v$ since the TRP AFR and SHL FT regimes have a similar distribution with almost identical medians and interquartile ranges (Fig. R3 right panel). But the governing physical processes of these two transport pathways are different: while the TRP AFR pathway is mainly affected by moist convective mixing and hence by microphysical processes, the SHL FT is primarily influenced by dry convective mixing without fractionation. Thanks to the characteristic isotope signal, a discrimination between these two fundamentally different transport pathways is possible.

The added value of stable water isotopes for distinguishing the TRP AFR and SHL FT pathways, which have a similar $q_v$ resulting, however, from different physical processes, can be also seen in Fig. 8b in the manuscript. The 60% frequency contours of the $\{q_v, \delta$D$\}$-pair distributions of the TRP AFR and SHL FT pathways overlap to a large extent along the $q_v$ axis but clearly differ in the $\delta$D direction. We probably did not stress this characteristic enough in the original manuscript. It is now addressed in more detail in the revised version:

*p. 20, l. 514–521: "The three African transport pathways also show some differences between their $\{q_v, \delta D\}$-pair distributions. There is a clear contrast in the $\delta D$ range of the SHL FT and TRP AFR distributions for the 60% contour, whereas the $q_v$ ranges largely overlap (Fig. 8b). Hence, these two transport pathways have a similar $q_v$ that apparently results from different physical processes. While the SHL FT pathway is primarily influenced by dry convective mixing without fractionation, the TRP AFR pathway is mainly affected by moist convection and thus by microphysical processes in addition to mixing. Thanks to the characteristic isotopic signature, a discrimination between these two fundamentally different transport pathways is possible. This emphasises the added value of water vapour isotopes for investigating physical processes and transport pathways that affect the subtropical tropospheric humidity."*

[Figure]

*Fig. R3: Boxplots of COSMO$_{iso}$ $\delta D$ (left panel, the same as Fig. 8a in the paper) and $q_v$ (right panel) for different categories of air parcels arriving at 500–700 hPa above Tenerife that originate from the upper-level extratropical North Atlantic (NA), from tropical Africa (TRP AFR), from the upper levels above the Saharan heat low (SHL FT) and from the Saharan heat low (SHL BL). The boxplots show the interquartile range by the extent of the box and the median by the black line in the box. The whiskers correspond to 1.5 times the proportion of the interquartile range past the lower and upper quartiles.*

– Fig. 9: Would it be insightful to show these trajectories in a $q_v$-$\delta D$ plot?

Thank you for the suggestion. We did the plot (Fig. R4 below) but decided to keep the original one in the paper, since displaying the temporal evolution of the air parcels in the $p$-$\delta D$ space is in our view better suited to explain the different transport pathways. In addition, we slightly changed Fig. 9: now the dots represent instantaneous values every 24 h instead of daily averaged values (original submission). The new way of visualising the trajectory data is more intuitive and facilitates the comparison with other figures.

[Figure]

*Fig. R4: $q_v$-$\delta D$ plot summarising the 10-day Lagrangian history of COSMO$_{iso}$ air parcels arriving at 500–700 hPa above Tenerife. The plot shows the median $q_v$ and $\delta D$ of the air parcels by the filled circles and the median p by the colours of the filled circles. The numbers indicate the days before the air parcels arrive over Tenerife, where red numbers represent air parcels originating from the boundary layer of the Saharan heat low (pathway SHL BL), purple numbers air parcels from the free troposphere above the Saharan heat low (SHL FT), green numbers air parcels from tropical Africa (TRP AFR), and blue numbers air parcels from the North Atlantic (NA).*

*– l827: Can't $q_v$ be similarly regarded as an integral measure of West African Dynamics?*

Since $q_v$ has limited capability to discriminate between air subsiding from the tropics (TRP AFR) and from the subtropics (SHL FT), we think that $\delta$D is a better measure of West African dynamics.

*→ I suggest to add some discussion about to what extent water isotopic observations provide an added value compared to $q_v$, or not.*

We hope that the above changes to the manuscript help convincing the reviewer and other readers of the paper that $\delta$D provides insights that could not be obtained by using $q_v$ alone.

**Minor Comments**

- *l 116: Blossey et al 2010 did not use a regional circulation model and did not study synoptic-scale variability. It was a cloud resolving model (higher resolution) in a very idealized setting that allows the simulation of the tropical circulation in a stationary state. I wouldn't list it with the other studies here.*

    This is true. We corrected the reference list accordingly.

- *l 228: Can you give more details about your cloud filtering? E.g., what threshold, what altitude for the cloud fraction? Is the fraction of selected scenes in the model similar to this fraction in reality? This could go in an appendix.*

    FTIR is always 100% cloud free, because the observations are controlled manually by the operating personnel of the FTIR spectrometer. We checked the cloud fraction at the Izaña observatory in the model for all times with FTIR observations and found that COSMO$_{iso}$ is also 100% cloud free. We changed the information about the cloud filtering in the manuscript accordingly.

    *p. 8, l. 228–230: "Finally, since the remote sensing retrieval processes only spectra measured in 100% cloud free conditions, we checked the cloud area fraction output from the model at all times when FTIR observations were available and found full consistency between the model and the observations."*

- *l 249: Same here: can you give more details about your cloud filtering? Why is it different from that for FTIR? What thresholds do you use? Is the fraction of selected scenes in the model similar to this fraction in reality? This could go in the same appendix as above.*

    The cloud filtering of the COSMO$_{iso}$ data for the IASI comparison differs from that for the FTIR comparison. This is mainly due to the different objectives of the two comparisons. While the comparison with FTIR aims at a direct comparison with individual observations (considering only model data with spatiotemporal colocation with the FTIR observations), the comparison with IASI is rather qualitative and aims at the analysis of the general atmospheric conditions around Tenerife (considering all model and satellite data within the chosen 10°x10° box around Tenerife, without considering more detailed colocation criteria between satellite and model data).

    Therefore, our intention is to detect and remove cloud-contaminated model data according to the characteristics of the IASI cloud filtering. For the IASI data, a strict cloud filter is applied, allowing no cloud contamination at all. Analogously, we only keep model data, where the vertically integrated variables liquid cloud water $q_c$ and ice cloud water $q_i$ (for grid-scale clouds) as well as total cloud cover (for subgrid-scale clouds) are (quasi) zero.

    The revised manuscript now includes a more detailed description of the cloud filtering:

    *p. 8, l. 247–255: "Since the MUSICA IASI retrievals provide results only for cloud-free scenes, a statistical cross-comparison of the MUSICA IASI dataset to COSMO$_{iso}$ simulations requires an analogous cloud filtering for the model data. For this purpose, we only consider model data where the*

*vertically integrated cloud water content $q_c$ and cloud ice content $q_i$ are zero. As these two conditions refer to grid-scale clouds, we remove sub-grid cloud fractions using the total cloud cover diagnostic (CLCT < 1e$^{-10}$). Afterwards, we multiply the COSMO$_{iso}$ water vapour isotope concentration profile with the simulated kernels, create the $\{q_v, \delta D\}$-pair product, and obtain a water vapour isotope concentration profile as would have been observed by IASI in the atmosphere simulated by COSMO$_{iso}$. These steps include an additional quality filtering according to the properties of the simulated averaging kernels, similar to the quality filtering of the MUSICA IASI $\{q_v, \delta D\}$-pair data (discussed in Diekmann et al., 2021b)."*

- l 252-253: What does this mean? Variability of what? Of the averaging kernels? Or of δD? Do you mean retrieval simulator combined with COSMOiso, or is it an intrinsic property of the simulator?

Thank you for pointing out this rather imprecise formulation. Following the approach from Schneider et al. (2017, see Fig. 5), we compared the actual MUSICA IASI averaging kernels with analogous retrieval simulations, resulting in a correlation between the simulated and actual MUSICA IASI averaging kernels of more than 95 %. This is documented in more detail in Diekmann (2021). We changed the manuscript as follows:

*p. 9, l. 255–257: "In its most recent version, the retrieval simulator achieves a correlation of more than 95 % between simulated and actual MUSICA IASI averaging kernels (compare to Fig. 5 in Schneider et al., 2017). The MUSICA IASI retrieval simulator is described in more detail in Diekmann (2021)."*

- l 323: I didn't follow why there are only seven trajectories. I thought there were seven days with many trajectories for each day?

The backward trajectories are started every hour in the period of 15 July to 30 August 2013 from every 20 hPa between 300 and 900 hPa above Tenerife. This results in 47 days with one trajectory per hour and vertical level. For the validation of the different transport pathways in COSMO$_{iso}$, we compare at each vertical level and within each 6-hour interval centred at the considered starting time the origin of all seven COSMO$_{iso}$ trajectories with the corresponding seven ERA-Interim trajectories. If at least four of the seven COSMO$_{iso}$ trajectories agree with the respective ERA-Interim trajectories about the origin, the transport in COSMO$_{iso}$ is considered "reliable" at the specific vertical level and time instance. We adapted the description of this part in the Lagrangian methods in section 2.4.

*p. 11, l. 326–332: "If at least four of the seven COSMO$_{iso}$ trajectories in the considered time window and at the specific vertical level agree with the corresponding ERA-Interim trajectories about the origin (continental Africa vs. North Atlantic), the transport in COSMO$_{iso}$ is considered "reliable" at the respective trajectory arrival time and vertical level.*

*For the comparison between COSMO$_{iso}$ and ground-based FTIR remote sensing observations, we regard a specific observation time as reliable if at least 75% of the levels between 400–700 hPa (representative for the 4.9 km FTIR retrieval level) are associated with a reliable transport according to the aforementioned evaluation criterion."*

- l 343-344: It must be clarified here that most of the apparent variability is associated with the vertical gradient in humidity and δD. "short-term variability" is confusing here because it evokes temporal variability, whereas here the vertical variability is probably dominant.

Thank you for pointing this out. It is true that the expression "short-term variability" is confusing in this context. We thus changed the sentence accordingly.

*p. 12, l. 351: "COSMO$_{iso}$ reasonably captures this observed short-term variability that is mostly associated with the vertical gradients in $q_v$ and $\delta D$."*

- l 345-347: "$\Delta\ln(q_v)$" → "$\Delta q_v$", since the values are in g/kg. The values for $\Delta\ln(q_v)$ would have no unit, or in %.

Changed according to the reviewer's suggestion.

- l 351: Does this sampling include vertical variations? If so, clarify that the shading probably mainly reflects the vertical variations?

No, this sampling only includes horizontal variations since it is performed in a purely horizontal box with no vertical extent. This guarantees that the shading only accounts for uncertainties in the synoptic-scale variability and does not reflect vertical variations.

- l 364: I don't understand this rationale. Why couldn't be the over-estimated $q_v$ and $\delta$D due to the dry and depleted tongue around 15°W that does not reach far enough towards the equator (around 30°N in COSMO and 25°N in ERAI)?

Thanks for this interesting question, which made us study this complex situation in greater detail. Indeed, backward trajectories calculated from COSMO_iso and ERA-Interim fields reveal that our original explanation was most likely not fully accurate. The flow situation on 30 July near the Canary Islands is rather complex and characterised by strong gradients. COSMO_iso correctly captures the transport of air from tropical West Africa (see Fig. R5 below) but strongly overestimates the specific humidity of the air arriving at 500 hPa in the area of the MUSICA aircraft campaign. The reason for this moist bias is a moistening by deep convection over tropical West Africa about five days prior to arrival, which might occur because of a too southerly location of the air parcels compared to ERA-Interim.

We changed the explanation in the revised manuscript accordingly:

*p. 12, l. 374–377: "Analysis of backward trajectories calculated from COSMO_iso and ERA-Interim fields (see Fig. A1 in the appendix) reveal that the overestimated $q_v$ and $\delta$D values in the model most likely result from a moistening of the air parcels by deep convection over tropical West Africa about five days prior to arrival over the Canaries and again near the African coast two days prior to arrival."*

[Figure]

*Fig. R5: (a) COSMO_iso and ERA-Interim backward trajectories started every hour between 10–13 UTC on 30 July 2013 (corresponding to observation period of airborne in situ measurements) from the 500 hPa level in the area of the MUSICA aircraft campaign (small red box). The trajectories are coloured according to their specific humidity $q_v$. The skewed black box depicts the COSMO_iso model domain in rotated coordinates. Dots and diamonds show the COSMO_iso (black) and ERA-Interim (grey) trajectory positions 2 and 5 days before arrival. (b) Temporal evolution of $q_v$ along the COSMO_iso and ERA-Interim backward trajectories displayed in (a). The red/blue solid lines represent the median of the 90 COSMO_iso /ERA-Interim trajectories, the red/blue dashed lines the corresponding interquartile ranges.*

Furthermore, we replaced Figs. A1–A7 in the appendix A by Fig. R5 and made the following changes in the text:

*p. 35, l. 893: "Appendix A: COSMO$_{iso}$ and ERA-Interim backward trajectories for 30 July 2013"*

*p. 35, l. 894–898: "In sect. 3.1, we argue that the overestimated $q_v$ and $\delta D$ values in COSMO$_{iso}$ on 30 July 2013 in the Canary Islands region most likely result from a moistening of the air by deep convection over tropical West Africa about 5 days prior to and again near the African coast 2 days prior to arrival. Figure A1 below demonstrates that COSMO$_{iso}$ correctly captures the transport of air from tropical West Africa but strongly overestimates the specific humidity of the air arriving at 500 hPa in the area of the MUSICA aircraft campaign."*

- l 411: "despite small" or "in spite of small"

Thank you, done.

- l 421: Couldn't there be an enriched bias for $\delta D$ observed by FTIR, and to a lesser extent by IASI? The comparison with in-situ data suggests that the model has an enriched bias, not a depleted bias. Previous studies cross-comparing different datasets have shown that the FTIR observations are often the most enriched (Risi et al., 2012b; Lacour et al., 2015).

Thank you for this useful comment. The inconsistency between the model comparison with aircraft in situ data and remote sensing data could be indeed interpreted as a positive $\delta D$ bias in the FTIR and IASI observations. However, a comparison between the different observational data sets is rather difficult since the in situ and remote sensing techniques sample the atmosphere differently in terms of spatial and temporal resolution. Aircraft in situ measurements may record very fine vertical and horizontal structures that cannot be resolved by the remote sensing retrieval process. Therefore, comparing the very short time series of airborne in situ measurements, which mainly reflects the vertical variability in $q_v$ and $\delta D$, with the two-months long time series of kernel-averaged remote sensing data, which primarily accounts for the temporal variability in $q_v$ and $\delta D$, may be misleading. More aircraft observations would be needed for a robust cross-comparison between the different observational datasets. Furthermore, we use the final MUSICA NDACC (FTIR) data product in this study, which is calibrated to the in situ aircraft profiles and therefore bias corrected (Schneider et al., 2016). The FTIR data used in the mentioned studies, however, is not bias corrected.

We added a short paragraph in the manuscript:

*p. 16, l. 437–442: "As a side note we mention that the inconsistency of the model comparison with aircraft in situ data vs. with remote sensing data might point to a positive $\delta D$ bias of the remote sensing observations. However, we use the final MUSICA NDACC (FTIR) data product in this study, which is calibrated to the in situ aircraft profiles and therefore bias corrected (Schneider et al., 2016). In addition, comparing the different observational data sets is difficult since the in situ and remote sensing techniques sample the atmosphere differently in terms of spatial and temporal resolution."*

Furthermore, we now introduce the alternative COSMO$_{iso}$ simulation with parameterised convection in Sect. 3.2 instead of Sect. 2.2. It is more intuitive to refer to the supplementary Fig. B1 after the discussion of the analogous Fig. 5 for the simulation with explicit convection.

*p. 17, l 443–447: "We briefly come back to the setup of our COSMO$_{iso}$ simulation with explicitly resolved convection. Figure B1 in the appendix shows an alternative simulation with parameterised convection, which leads to larger and mostly positive model biases in comparison with airborne, ground- and space-based observations. This is in agreement with previous studies, which have already reported positive mid-tropospheric $\delta D$ biases in model simulations with parameterised convection due to an overestimated vertical moisture transport (e.g., Werner et al., 2011; Risi et al., 2012a, b; Christner et al., 2018)."*

- Are the COSMOiso simulations and observations used in the comparison available in a repository?

COSMO$_{iso}$ output data are available from the authors upon request (fabienne.dahinden@env.ethz.ch). Airborne in situ and ground-based FTIR remote sensing observations can be accessed via https://www.imk-asf.kit.edu/english/musica-data.php. Space-based IASI observations for July and August 2013 are available upon request (matthias.schneider@kit.edu).

At the end of the paper, there is now a short paragraph about the data availability.

**References**

[revised manuscript text omitted]

---

## Author Response (AR2)

**Answers to the editor's comments**

We thank the editor for carefully reading our revised manuscript and the author's final comments, and for his additional comments. Below, we address each comment point by point. The comments are repeated in blue, our responses are given in black, and the changes to the manuscript in black italic.

1) I. 251: Please change to scientific power notation 10-10.

Changed according to the editor's suggestion

2) In line with ACPs data policy, please make the COSMOiso output and the IASI data available in a permanent repository with a doi linked to the paper.

The COSMOiso output fields are now published with the doi https://doi.org/10.3929/ethz-b-000506055 and the IASI data is accessible with the doi https://dx.doi.org/10.35097/492. We changed the data availability statement accordingly in the manuscript:

p. 37, I. 920–923: "The COSMOiso simulation output is published on the ETH research collection with the doi https://doi.org/10.3929/ethz-b-000506055 (Dahinden et al., 2021). Airborne in situ and ground-based FTIR remote sensing observations can be accessed via https://www.imk-asf.kit.edu/english/musica-data.php. Space-based IASI observations for July and August 2013 are published on https://dx.doi.org/10.35097/492 (Diekmann et al. 2021c)."

3) Please clarify whether the COSMOiso code is published and from where it can be obtained.

The availability of the COSMOiso code and the Fortran code for the trajectory calculations is now specified in the manuscript:

p. 37, I 923–929: "The particular version of the COSMO model used in this study is based on the official version 4.18 with additionally implemented stable water isotope physics and is available under license (see http://www.cosmo-model.org/content/consortium/licencing.htm for more information, last access: 20 September 2021). COSMO may be used for operational and for research applications by the members of COSMO. Moreover, within a license agreement, the COSMO model may be used for operational and research applications by other national (hydro-)meteorological services, universities, and research institutes. The Fortran code for the trajectory calculations is available under http://iacweb.ethz.ch/staff/sprenger/lagranto/download.html."

**References**

Dahinden, F., Aemisegger, F., Pfahl, S., and Wernli, H.: Numerical weather simulation using COSMOiso over the eastern subtropical North Atlantic in July and August 2013, Research Collection, ETH Zurich, Zurich, https://doi.org/10.3929/ethz-b-000506055, 2021.

Diekmann, C. J., Schneider, M., Ertl, B.: Data for "Disentangling different moisture transport pathways over the eastern subtropical North Atlantic using multi-platform isotope observations and high-resolution numerical modelling". Institute of Meteorology and Climate Research, Atmospheric Trace Gases and Remote Sensing (IMK-ASF), Karlsruhe Institute of Technology (KIT), https://dx.doi.org/10.35097/492, 2021c.